# Aldolase-regulated G3BP1/2⁺ condensates control insulin mRNA storage in beta cells

Esteban Quezada [1,2,3], Klaus-Peter Knoch[1,2,3], Jovana Vasiljevic[1,2,3], Annika Seiler[1,2,3], Akshaye Pal[4], Abishek Gunasekaran [1,2,3], Carla Münster[1,2,3], Daniela Friedland[1,2,3], Eyke Schöniger[1,2,3], Anke Sönmez[1,2,3], Pascal Roch[1,2,3], Carolin Wegbrod[1,2,3], Katharina Ganß[1,2,3], Nicole Kipke[1,2,3], Simon Alberti [5], Rita Nano[6,7], Lorenzo Piemonti[6,7], Daniela Aust[8], Jürgen Weitz[2,3,9], Marius Distler[2,3,9] & Michele Solimena [1,2,3✉]

## Abstract

Upregulation of insulin mRNA translation upon hyperglycemia in pancreatic islet β-cells involves several RNA-binding proteins. Here, we found that G3BP1, a stress granule marker downregulated in islets of subjects with type 2 diabetes, binds to insulin mRNA in glucose concentration-dependent manner. We show in mouse insulinoma MIN6-K8 cells exposed to fasting glucose levels that G3BP1 and its paralog G3BP2 colocalize to cytosolic condensates with eIF3b, phospho-AMPKα$^{Thr172}$ and Ins1/2 mRNA. Glucose stimulation dissolves G3BP1⁺/2⁺ condensates with cytosolic redistribution of their components. The aldolase inhibitor aldometanib prevents the glucose- and pyruvate-induced dissolution of G3BP1⁺/2⁺ condensates, increases phospho-AMPKα$^{Thr172}$ levels and reduces those of phospho-mTOR$^{Ser2448}$. G3BP1 or G3BP2 depletion precludes condensate assembly. KO of G3BP1 decreases Ins1/2 mRNA abundance and translation as well as proinsulin levels, and impaires glucose-stimulated insulin secretion. Further, other insulin secretagogues such as exendin-4 and palmitate, but not high KCl, prompts the dissolution of G3BP1⁺/2⁺ condensates. G3BP1⁺/2⁺/Ins mRNA⁺ condensates are also found in primary mouse and human β-cells. Hence, G3BP1⁺/2⁺ condensates represent a conserved glycolysis/aldolase-regulated compartment for the physiological storage and protection of insulin mRNA in resting β-cells.

**Keywords** Stress Granules; Insulin; Translation; Islet; Diabetes
**Subject Categories** Organelles; RNA Biology; Translation & Protein Quality

## Introduction

Pancreatic islet beta cells play a major role in maintaining glucose homeostasis by synthesizing and secreting the peptide hormone insulin in response to elevation of blood glucose levels (Karpińska and Czauderna, 2022). Insulin, in turn, lowers glycemia through insulin receptor signaling, hence promoting glucose uptake into muscle cells and adipocytes while inhibiting glycogenolysis and gluconeogenesis in hepatocytes. Accordingly, impaired beta cell function together with elevated insulin resistance cause hyperglycemia and type 2 diabetes (T2D) (Zheng et al, 2018). Beta cell dysfunction in T2D has been attributed to several mechanisms, often combined with each other, including reduced oxidative phosphorylation (Supale et al, 2012; Dludla et al, 2023), prolonged ER stress (Back and Kaufman, 2012; Mustapha et al, 2021; Shrestha, 2021), chronic islet inflammation (Donath et al, 2003; Donath et al, 2013; Kulkarni et al, 2022; Rohm et al, 2022) and amyloidosis (Kahn et al, 1999; Jurgens et al, 2011) resulting in beta cell dedifferentiation (Talchai et al, 2012; Son and Accili, 2023) and death (Butler et al, 2003). Yet, it is important to appreciate that a clear explanation for how beta cell dysfunction evolves during the progression from normoglycemia to T2D is still missing. Also, it is conceivable that the contribution of different mechanisms to altered insulin secretion varies among different individuals, as suggested by the identification of several disease clusters (Ahlqvist et al, 2018). Recent evidence, in particular, suggests that a lower threshold for glucose-stimulated insulin release can set in motion the vicious circle between insulin resistance and hyperinsulinemia, leading eventually to beta cell decompensation and T2D (Cohrs et al, 2020; Johnson, 2021).

To gain further insight into the physiology of beta cells and its pathophysiology, we undertook the systematic transcriptomic analysis of pancreatic islets retrieved by laser capture microdissection from surgical specimens of metabolically profiled living donors who underwent pancreatectomy for a variety of pancreatic

[1]Molecular Diabetology, University Hospital and Faculty of Medicine Carl Gustav Carus, TU Dresden, Dresden, Germany. [2]Paul Langerhans Institute Dresden (PLID) of the Helmholtz Center Munich at the University Hospital and Faculty of Medicine Carl Gustav Carus, TU Dresden, Dresden, Germany. [3]German Center for Diabetes Research (DZD e.V.), Neuherberg, Germany. [4]Max Planck Institute of Molecular Cell Biology and Genetics, 01307 Dresden, Germany. [5]Biotechnology Center (BIOTEC), Center for Molecular and Cellular Bioengineering, TU Dresden, Dresden, Germany. [6]Diabetes Research Institute, IRCCS Ospedale San Raffaele, Milan, Italy. [7]Università Vita-Salute San Raffaele, Milan, Italy. [8]Department of Pathology, University Hospital and Faculty of Medicine Carl Gustav Carus, TU Dresden, Dresden Germany, TU Dresden, Dresden, Germany. [9]Department of Visceral, Thoracic and Vascular Surgery, University Hospital and Faculty of Medicine Carl Gustav Carus, TU Dresden, Dresden, Germany. ✉E-mail: michele.solimena@tu-dresden.de

disorders (Solimena et al, 2018; Barovic et al, 2019). These investigations identified the RNA-binding protein G3BP1 to be among the most significantly downregulated genes in islets of living donors with T2D compared to normoglycemic individuals (Wigger et al, 2021). G3BP1 and its paralog G3BP2, also known as Ras-GTPase Activating Protein SH3 domain-Binding Proteins 1 and 2, have been shown to be essential components of the stress granules (Guillén-Boixet et al, 2020; Yang et al, 2020; Kang et al, 2021). These non-membranous cytosolic condensates, which result from the interaction of several RNA-binding proteins (RBPs) with polyadenylated mRNAs, form in cells with arrested translation under various stress conditions, such as oxidative stress, nutrient deprivation, heat, and UV radiation (Mahboubi and Stochaj, 2017). Upon stressor removal, stress granules disassemble, allowing mRNAs to re-engage with ribosomes for protein synthesis (Wheeler et al, 2016). Stress granule assembly may also serve as a cellular defense mechanism in the early stages of viral infection by suspending translation (Jayabalan et al, 2021). Accordingly, some viruses have evolved strategies to disrupt stress granule functionality during later infection stages (Lloyd 2012; Yang et al, 2018). Besides G3BP1/2, other constituents of stress granules include RNA-binding proteins TIA-1/R and PABP1, the small 40S ribosomal subunit, and translation initiation factors eIF2, eIF3, eIF4A, eIF4G and eIF4E, although the composition of these condensates can vary depending on the cell type and the specific stress encountered (Moutaoufik et al, 2014; Protter and Parker 2016; Fay and Anderson, 2018).

Acute elevation of beta cell insulin production in response to stimulation with glucose, but also exendin-4, does not depend on insulin gene transcription, but on the increased translation of pre-existing copies on insulin mRNA (Itoh and Okamoto, 1980; Welsh et al, 1985; Knoch et al, 2004; Knoch et al, 2006). However, little is known about where beta cells store insulin mRNA in fasting conditions. In the present study we began to fill this gap of knowledge by investigating the possible relationship between stress granules with insulin mRNA storage sites in beta cells, as a necessary premise to understand the potential implications of downregulated gene expression of *G3BP1* in islets of subjects with T2D.

# Results

## *Ins1/2* mRNA is enriched in G3BP1$^+$/G3BP2$^+$/eIF3b$^+$ condensates in resting MIN6-K8 cells

Our studies began by analyzing the expression of G3BP1 and its paralogue G3BP2 in glucose-responsive mouse insulinoma MIN6-K8 cells. In these cells, the mRNA levels of *G3BP1*, *G3BP2*, as well as of *Ins1* and *Ins2* mRNAs did not change upon elevation of extracellular glucose from 2.8 mM (resting condition) to 16.7 mM (stimulating condition) for 30 min. (Fig. 1A). Likewise, the protein levels of G3BP1 and G3BP2 remained unchanged (Fig. 1B). Confocal microscopy showed that in resting cells G3BP1 and G3BP2 were enriched in cytosolic condensates, potentially corresponding to G3BP1$^+$ and G3BP2$^+$ stress granules described in other cell types (Mahboubi and Stochaj, 2017). Such compartments typically form in response to conditions eliciting pronounced energy depletion, such as starvation or exposure to

oxidative stress-inducing agents, to retain mRNAs while suppressing their translation, thereby reducing energy expenditure. Consistent with this possibility, G3BP1$^+$/G3BP2$^+$ condensates were also enriched in eIF3b, which is part of the small ribosomal unit and also a marker of stress granules (Yang et al, 2020) (Fig. 1C). Notably, in MIN6-K8 cells stimulated with 16.7 mM glucose, G3BP1$^+$/G3BP2$^+$/eIF3b$^+$ were redistributed throughout the cytosol and their condensates were no longer detectable (Fig. 1D). Since stress granules also contain mRNA strands attached to RBPs and to the 40S small ribosomal subunit (Wolozin and Ivanov, 2019), we examined the distribution of *Ins1/2* mRNA and *LacZ* mRNA as a negative control by single molecule RNA FISH (smRNA FISH). We observed that *Ins1/2* mRNA was confined to G3BP1/2$^+$ condensates in resting conditions (Fig. 1E). Upon stimulation with 16.7 mM glucose, all signals, including that for *Ins1/2* mRNA, diffused throughout the cytosol (Fig. 1F). We further addressed the distribution of mRNAs encoding for other cargoes of the insulin secretory granules, such as *Ptprn/ICA512*, *Pcsk1* and *Pcsk2*, as their translation is also rapidly upregulated in response to elevation of glucose levels and incretin stimulation to enable the biogenesis of these organelles (Knoch et al, 2004; Knoch et al, 2006). However, none of them was detectable in G3BP1$^+$ condensates of resting MIN6-K8 cells (Fig. EV1).

## G3BP1$^+$ condensates in MIN6-K8 cells occur in conditions distinct from those associated with the biogenesis of arsenate-induced stress granules

In view of their glucose-regulated dynamics we reasoned that the G3BP1$^+$ condensates observed in insulin-producing cells could reflect a physiological rather than a response to severe stress. To test this hypothesis, we examined the occurrence of G3BP1$^+$ condensates in resting or glucose stimulated MIN6-K8 cells in the presence or absence of a strong oxidative agent such as 1 mM sodium arsenate, which elicits the formation of canonical stress granules in other cell types. As expected, glucose stimulation reduced the number of G3BP1$^+$/eIF3b$^+$/*Ins1/2* mRNA$^+$ condensates/cell compared to resting cells (Fig. 2A–C). Upon treatment with 1 mM sodium arsenate the number of G3BP1$^+$/eIF3b$^+$/*Ins1/2* mRNA$^+$ condensates/cell as well as their total area was increased regardless of the glucose levels (Fig. 2A–C), while the size of individual condensates was unchanged (Fig. 2D). As expected (Buchan et al, 2008), in cells kept at rest with 2.8 mM glucose G3BP1$^+$ condensates were resolved by the addition of 100 µg/ml of cycloheximide, which blocks elongation of translation by stabilizing mRNA-ribosome complexes (Fig. 2A–D).

In different cell types the induction of stress granules in response to various stressors, such as sodium arsenate, involves the phosphorylation of eIF2α. On the other hand, in resting and glucose-stimulated MIN6-K8 cells treated with sodium arsenate, the levels of phospho-eIF2α were similarly elevated compared to MIN6-K8 cells exposed to low or high glucose levels alone (Fig. 2E,F). Formation of stress granules has also been associated with phosphorylation of the nutrient-sensor AMPKα on threonine 172, and the localization of phospho-AMPKα$^{Thr172}$ to these condensates (Mahboubi and Stochaj, 2017). We observed a significant reduction of phospho-AMPKα$^{Thr172}$ levels in glucose-stimulated MIN6-K8 cells compared to resting cells (Fig. 2E,F). Unexpectedly, however, treatment of resting cells with 1 mM sodium arsenate correlated with lower rather than higher levels

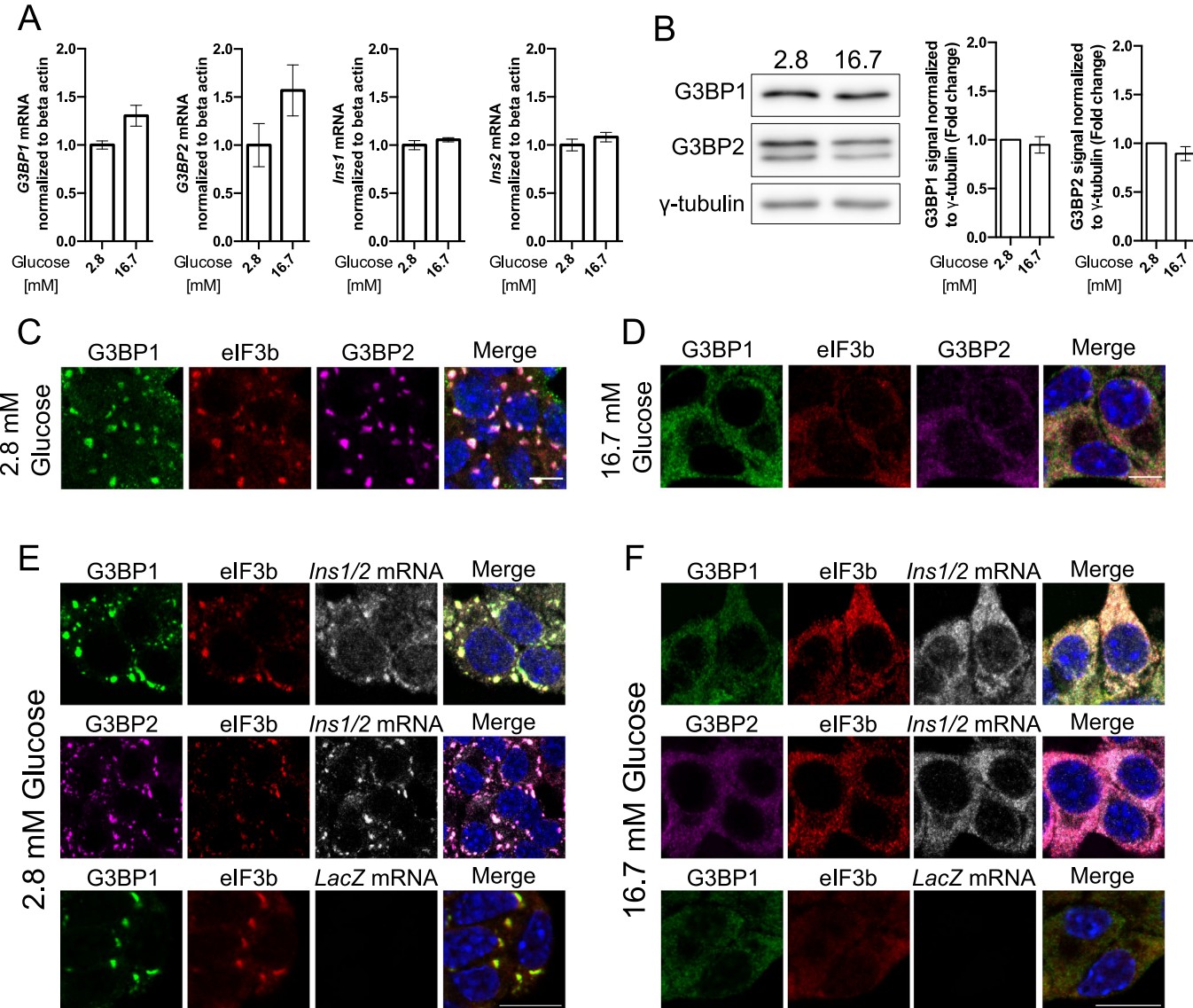

**Figure 1. Co-localization of G3BP1, G3BP2, and eIF3b with *Ins1/2* mRNA in glucose-regulated cytosolic condensates in MIN6-K8 cells.**

(A) mRNA levels of *G3BP1*, *G3BP2*, *Ins1*, and *Ins2* in resting (2.8 mM) and stimulating (16.7 mM) glucose concentrations as measured by qRT-PCR. (B) Protein levels of G3BP1 and G3BP2 under resting and stimulating glucose concentrations as assessed by western blot. (C, D) Immunostaining for G3BP1 (green), G3BP2 (magenta), eIF3b (red) in resting or stimulating glucose concentrations. Nuclei are stained with DAPI (blue). (E, F) Immunostaining for G3BP1 (green), G3BP2 (magenta), eIF3b (red), and smRNA FISH for *Ins1/2* or *LacZ* (gray) in resting glucose concentrations. Nuclei are stained with DAPI (blue). Data information: The intensity of the bands in the western blots (B) was measured in arbitrary units using ImageStudioLite software, normalized to the γ-tubulin loading control, and fold change was calculated relative to the resting glucose condition. The values represent the mean ± SD from three independent experiments and were analyzed via a paired *t*-test with Mann–Whitney correction. For qRT-PCR, 3 technical replicates and for western blot 1 technical replicate of each condition per independent experiment were performed. Scale bar in (C, D) = 5 μm and in (E, F) = 10 μm. Source data are available online for this figure.

of phospho-AMPKα$^{Thr172}$ relative to cells exposed to 2.8 mM glucose only (Fig. 2E,F), while in the case of glucose-stimulated cells sodium arsenate treatment correlated with increased phospho-AMPKα$^{Thr172}$ levels relative to cells treated with high glucose alone. Moreover, in resting cells phospho-AMPKα$^{Thr172}$ was detected in G3BP1$^+$ condensates, while its signal redistributed together with those of G3BP1 and eIF3b in glucose stimulated cells (Fig. 2G). Studies in other cell types suggested that the assembly of stress granules is regulated through the phosphorylation of G3BP1 on serines 149 and 232 within

its acidic intrinsically disordered region (Reineke et al, 2017; Guillén-Boixet et al, 2020; Tourrière et al, 2023). However, in the case of MIN6-K8 cells, glucose stimulation did not enhance the levels of phospho-G3BP1$^{S149}$ and phospho-G3BP1$^{S232}$, either individually or cumulatively (Figs. 2H–J and EV2A–D). Collectively, these results suggest that in MIN6-K8 cells G3BP1$^+$/eIF3b$^+$/*Ins1/2* mRNA$^+$ condensates resemble those induced by sodium arsenate, although in resting cells exposed to this treatment phospho-AMPKα$^{Thr172}$ levels were lower than in cells kept in low glucose only (Panas et al, 2019; Tourrière and Tazi, 2019).

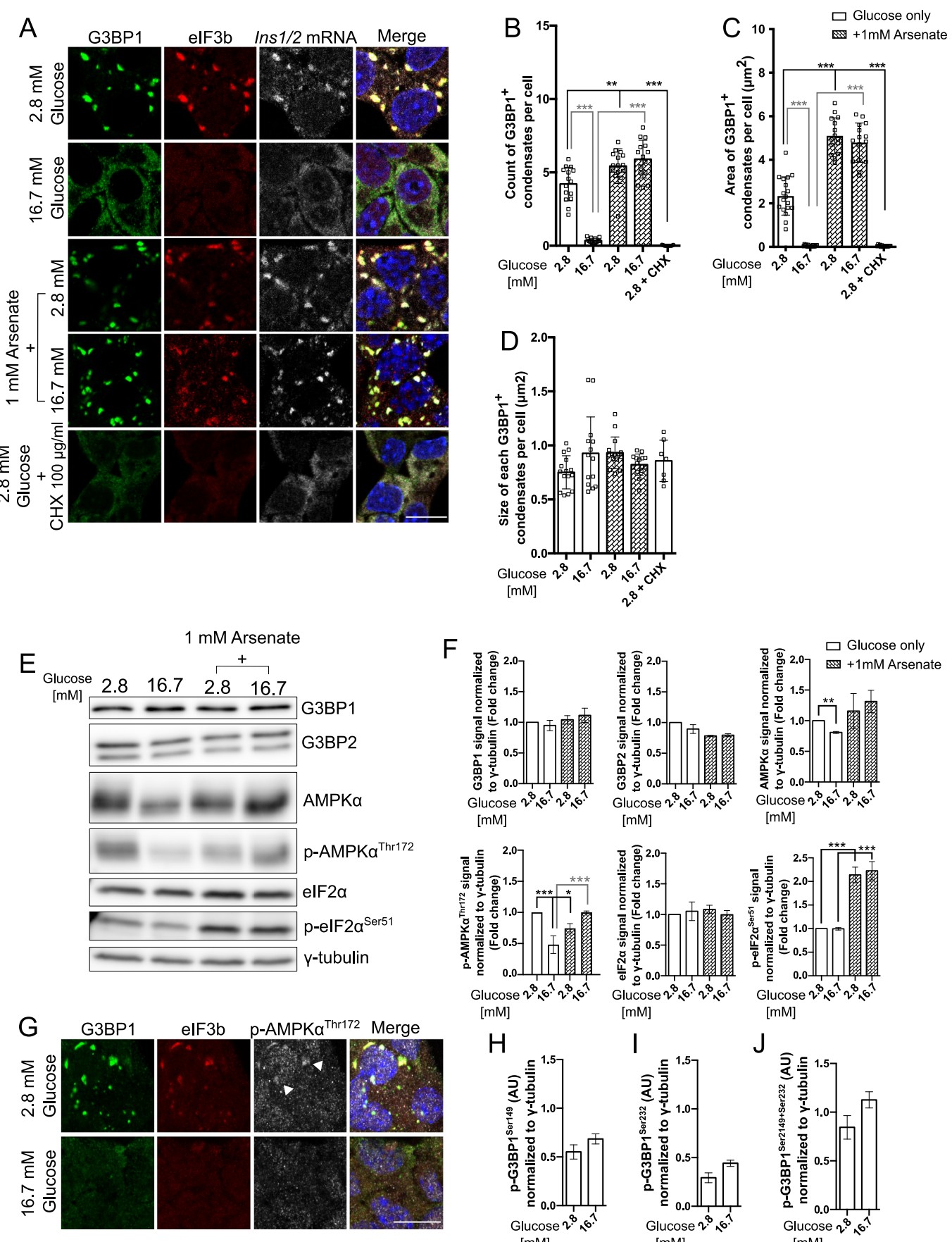

**Figure 2.  Differential behavior and signaling pathways of G3BP1⁺ condensates in response to glucose stimulation and oxidative stress in MIN6-K8 cells.**

(A) Immunostaining for G3BP1 (green), eIF3b (red), and smRNA FISH for *Ins1/2* (gray) under resting and stimulating glucose conditions with and without 1 mM sodium arsenate. Nuclei are stained with DAPI (blue). (B) Count of G3BP1⁺ condensates per cell in all tested conditions as in (A). (C) Area of G3BP1⁺ condensates per cell ($\mu m^2$) in all tested conditions as in (A). (D) Size of each G3BP1⁺ condensate ($\mu m^2$) in all tested conditions as in (A). (E, F) Western blot (E) and quantifications (F) for G3BP1, G3BP2, AMPKα, phospho-AMPKα, eIF2α, phospho-eIF2α and gamma-tubulin across varying conditions: resting and stimulating glucose concentrations, with and without 1 mM sodium arsenate. (G) Immunostaining for G3BP1 (green), eIF3b (red) and Phospho-AMPKα (gray) under resting or stimulating glucose conditions. White arrows indicate P-AMPKα signal in G3BP1⁺ condensates. Nuclei are stained with DAPI (blue). (H–J) Quantification of phospho-G3BP1-S149 and phospho-G3BP1-S232 alone or combined based on the Western blots shown in Fig. EV2 for the corresponding phospho-G3BP1 species. Data information: The intensity of the bands in the Western blots was quantified in arbitrary units utilizing ImageStudioLite software, normalized against the γ-tubulin loading control, with fold changes for each condition calculated relative solely to the resting glucose condition. Presented values denote the mean ± SD derived from three independent experiments, analyzed via one-way ANOVA. Values with *$p < 0.05$, **$p < 0.01$ and ***$p < 0.001$ were considered statistically significant relative to the conditions as shown in each graph. Quantifications from (B–D) were taken from at least 15 images (at least 5 per independent experiment). Western blot quantifications correspond to 1 technical replicate of each condition per independent experiment. Scale bar = 5 μm. (B): 2.8 vs 2.8 + Arsenate ($p = 0.000485$); 2.8 + CHX ($p < 0.0001$). 16.7 vs 2.8 ($p < 0.0001$). 16.7 + Arsenate ($p < 0.0001$). (C): Every comparison $p < 0.0001$. (F): AMPKα graph 2.8 vs 16.7 ($p = 0.005$). phospho-AMPKα graph 2.8 vs 16.7 ($p = 0.0003$); 2.8 + Arsenate ($p = 0.0221$). 16.7 vs 16.7 + Arsenate ($p = 0.0003$). phospho-eIF2α graph both comparison $p < 0.0001$. Source data are available online for this figure.

## Glucose stimulated G3BP1⁺ condensate dissolution is aldolase and ATP-dependent

In liver cells the key glycolytic enzyme aldolase regulates phospho-AMPKα^Thr172 levels through the recruitment of the cytosolic AXIN/LKB1 complex to the lysosomal membranes when not bound to its substrate fructose 1,6-biphosphate (FBP), i.e., in conditions of low glucose (Lin and Hardie, 2018). Conversely, increased glycolysis and FBP-aldolase levels prompt the dissociation of the AXIN/LKB1 complex from the lysosomes, with reduction of phospho-AMPKα^Thr172 levels and concomitant elevation of those of activated phospho-mTOR^Ser2448. It is also known that stress granule dynamics are modulated by ATP (Jain et al, 2016; Wheeler et al, 2016). To test whether aldolase can regulate G3BP1⁺ condensates through phospho-AMPKα^Thr172 in insulin-producing cells, we treated MIN6-K8 cells with the aldolase specific inhibitor aldometanib (Zhang et al, 2022). As expected, ATP levels were increased upon cell exposure to 16.7 mM glucose and reduced upon mitochondrial inhibition with rotenone (Fig. 3A). ATP levels were also increased upon cell incubation with 20 mM pyruvate, which lies downstream of the aldolase product glyceraldehyde-3-phosphate along the glycolytic pathway and can enter MIN6-K8 cells, albeit not primary beta cells, due to their expression of MCT-4 (Fig. EV3G). Remarkably, treatment with 200 nM aldometanib reduced ATP levels not only in resting or glucose-stimulated cells, but also in those co-treated with pyruvate, suggesting that at least in these cells aldolase's control on energy supply extends beyond its role in glycolysis. Accordingly, aldometanib treatment increased in a dose-dependent fashion the levels of phospho-AMPKα^Thr172 while reducing those of phospho-mTOR^Ser2448 (Fig. EV3A–C). Furthermore, it prevented the otherwise pyruvate-induced dissolution of G3BP1⁺/eIF3b⁺/*Ins1/2* mRNA⁺ condensates in cells incubated with either 2.8 or 16.7 mM glucose (Fig. 3B,C). Specifically, in aldometanib treated cells condensates were increased in number and total area, while their individual size was unchanged (Fig. 3D–E).

## G3BP1, but not G3BP2, is required for *Ins1/2* mRNA stability

G3BP1 and G3BP2 are critical constituents of stress granules. In some instances, the occurrence of stress granules requires the expression of both G3BP1 and G3BP2 (Kedersha et al, 2016; Hofmann et al, 2021), while in other cases absence of only one of these factors prevents it (Yang et al, 2020). To investigate whether G3BP1 and G3BP2 are essential components of *Ins1/2* mRNA⁺

condensates, we characterized *G3BP1⁻ᐟ⁻* and *G3BP2⁻ᐟ⁻* MIN6-K8 cells generated by CRISPR/Cas9 technology (Figs. 4A,B and EV6). Lack of either G3BP1 and G3BP2 alone was sufficient to abolish the presence of eIF3b⁺/*Ins1/2* mRNA⁺ condensates in resting cells, with the signal for both molecules being more evenly distributed throughout the cytosol (Figs. 4C and EV4A). In the case of resting *G3BP2⁻ᐟ⁻* cells, G3BP1 was also more diffused compared to control cells, albeit still enriched in particles of undefined identity (Fig. 4C). Conversely, in resting *G3BP1⁻ᐟ⁻* cells, G3BP2 was more homogeneously distributed in the cytoplasm (Fig. EV4A). The presence of glucose-regulated condensates in *G3BP1⁻* cells under resting glucose could be restored by expression of mCherry-hG3BP1 (Fig. 4D). Notably, though, mCherry-hG3BP1⁺ condensates were not effectively resolved by glucose stimulation, possibly due to interference of the N-terminal tagging of the protein (Fig. EV4B). In *G3BP1⁻ᐟ⁻* cells the lack of the condensates was associated with reduced levels of both insulin transcripts, and especially *Ins1* mRNA, regardless of whether the cells were resting (Fig. 4E) or had been stimulated with 25 mM glucose (Fig. EV5A). In *G3BP2⁻ᐟ⁻* cells *Ins1* mRNA levels were instead unchanged relative to wild-type cells, while those of *Ins2* mRNA were elevated, independently of the glucose concentration (Figs. 4E and EV5A). Treatment for up to 24 h with 5 μg/ml Actinomycin D, which blocks transcription, reduced *Ins1* mRNA content in *G3BP1⁻ᐟ⁻*, but not in *G3BP2⁻ᐟ⁻* cells, indicating a potential role of G3BP2 in *Ins1* mRNA degradation (Fig. 4F). Moreover, 24 h treatment with 5 μg/ml Actinomycin D correlated with lower levels of *Ins2* mRNA in *G3BP1⁻ᐟ⁻*, but not in MIN6-K8 WT and *G3BP2⁻ᐟ⁻* cells (Fig. 4G), corroborating the possibility that G3BP1 exerts a protective/stabilizing role for *Ins1/2* transcripts. By expressing mCherry-hG3BP1 in *G3BP1⁻ᐟ⁻* cells, the expression levels of *Ins1* mRNA were partially rescued to ~50% compared to WT cells (Fig. 4H), while those of *Ins2* mRNA were approximately doubled, suggesting an increased stability of this transcript (Fig. 4H). At the protein level, proinsulin content was lower in *G3BP1⁻ᐟ⁻* cells in both resting and glucose-stimulated conditions compared to wild-type cells, although the amount of insulin was unchanged (Fig. 4I). Conversely, in *G3BP2⁻ᐟ⁻* cells neither proinsulin nor insulin content were altered compared to wild-type cells (Fig. 4I). Similar findings were obtained with more accurate and sensitive measurements for proinsulin and insulin by ELISA and HTRF, respectively (Fig. EV5B,C). Proinsulin content could be restored in *G3BP1⁻ᐟ⁻* cells expressing mCherry-G3BP1 (Fig. 4J). Both *G3BP1⁻ᐟ⁻* and

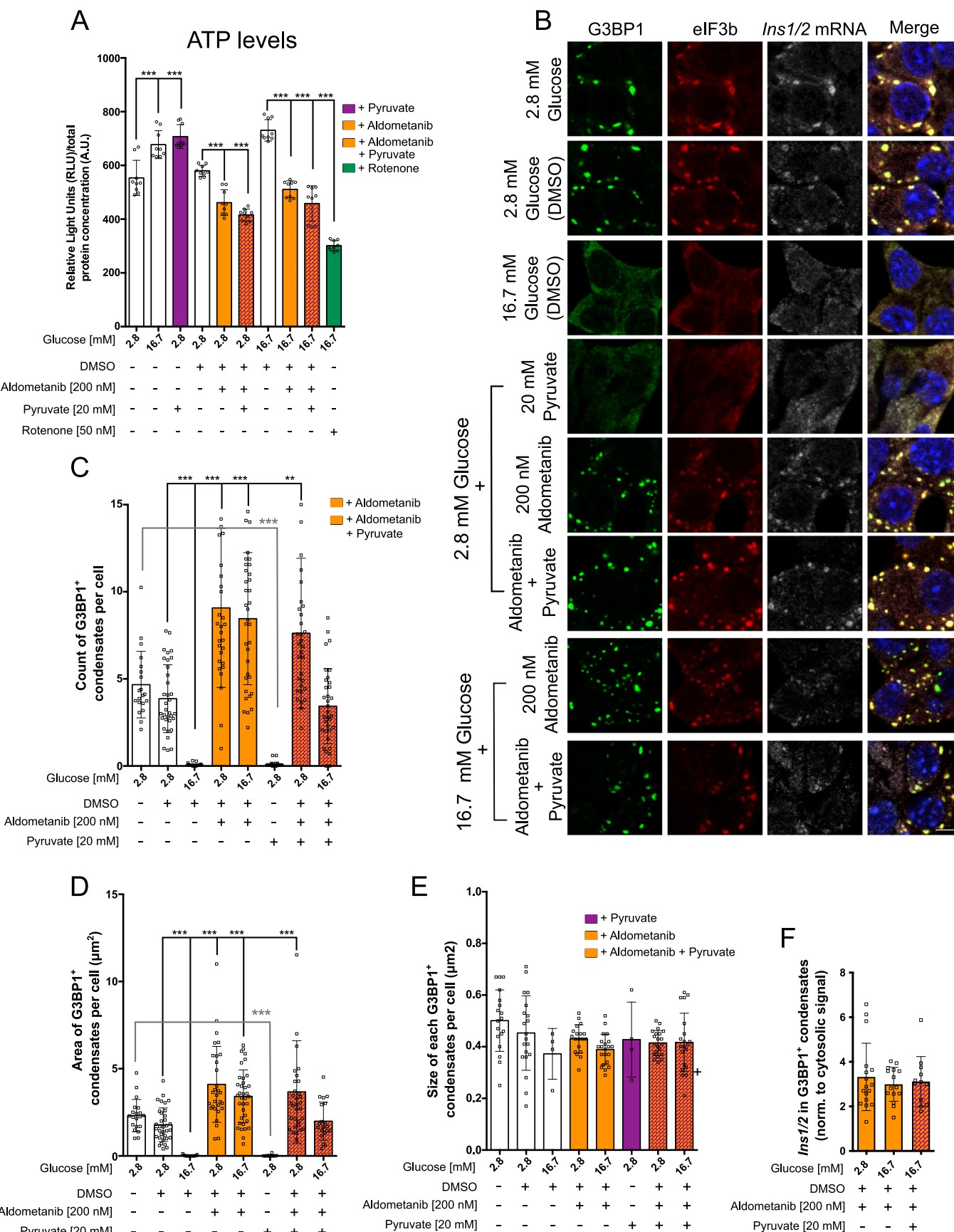

**Figure 3. G3BP1⁺ condensates dissolution is ATP- and aldolase-dependent.**

(A) Luciferase-based assay for ATP measurement. Graph shows values of relative light units (RLU) normalized to total protein levels for each condition. Conditions are under resting or glucose stimulating conditions with the addition of DMSO (vehicle), Aldometanib (200 nM), pyruvate (20 mM) or rotenone (50 nM). (B) Immunostaining for G3BP1 (green), eIF3b (red) and smRNA FISH for *Ins1/2* (gray) under resting or stimulating glucose conditions the addition of DMSO (vehicle), Aldometanib (200 nM) or pyruvate (20 mM). Nuclei are stained with DAPI (blue). (C) Count of G3BP1⁺ condensates per cell in all tested conditions as in (B). (D) Area of G3BP1⁺ condensates per cell (μm²) in all tested conditions as in (B). (E) Size of each G3BP1⁺ condensate (μm²) in all tested conditions as in (B). (F) Quantification of *Ins1/2* mRNA fluorescence intensity in G3BP1⁺ condensates normalized to the corresponding cytosolic signal in the conditions indicated in the graph. Data information: Presented values denote the mean ± SD derived from three independent experiments, analyzed via one-way ANOVA. Values with *$p < 0.05$, **$p < 0.01$, and ***$p < 0.001$ were considered statistically significant relative to the conditions as shown in each graph. Quantifications of A were from 3 replicates for each independent experiment. Quantifications from (C–E) were taken from at least 15 images (at least 5 per independent experiment). Scale bar = 10 μm. (A): Every comparison $p < 0.0001$. (C): Every comparison $p < 0.0001$, except for 2.8 DMSO vs 2.8 + aldometanib + pyruvate ($p = 0.0023$). (D): 2.8 vs 2.8 + pyruvate ($p = 0.00018$). 2.8 DMSO vs 16.7 DMSO ($p = 0.0007$); 2.8 + aldometanib ($p < 0.0001$); 16.7 + aldometanib ($p = 0.0008$); 2.8 + aldometanib + pyruvate ($p = 0.0001$). Source data are available online for this figure.

*G3BP2⁻ᐟ⁻* cells showed impaired glucose-stimulated insulin secretion with a reduced stimulation index compared to WT MIN6-K8 cells (Fig. EV5E,G), with *G3BP1⁻ᐟ⁻* cells displaying also a higher proinsulin release in resting conditions (Fig. EV5D), which is a hallmark of beta cell dysfunction in type 2 diabetes. Other cargoes, such as *PTRPN* and *Pcsk2*, showed higher transcript levels in *G3BP2⁻ᐟ⁻* cells, but not in protein levels compared to wild type. However, *Pcsk1* transcript and protein levels were higher in *G3BP1⁻ᐟ⁻* cells compared to wild type (Fig. EV5H–M). Suggesting distinct regulatory mechanisms for different insulin secretory granule cargoes.

It has been shown that G3BP1 plays a role in polysome turnover (Meyer et al, 2020). In addition, preproinsulin biosynthesis has been described to occur in polysomes (Wolin and Walter 1993). To ascertain if this could account for lower proinsulin levels in *G3BP1⁻ᐟ⁻* cells, a polysomal profiling was conducted using a 10–50% sucrose gradient. This analysis revealed that wild-type cells responded aptly to varying glucose levels, containing a pronounced monosomal peak (80S) in resting conditions (Fig. 4K), which decreased by half upon glucose stimulation, when polysome peaks were enriched (Fig. 4K). This suggests that in stimulating conditions the cells transition to active protein translation, as evidenced by the enrichment of polysomal subpopulations peaks. Instead, in *G3BP1⁻ᐟ⁻* cells only the first two polysomal subpopulation peaks were present (Fig. 4K), while upon glucose stimulation the height of the 80S peak was twice the size relative to wild-type cells (Fig. 4K). This profile suggests that absence of G3BP1 does not promote translation of polysome associated transcripts, as monosomes cannot transition into polysomes. The polysomal profile of *G3BP2⁻ᐟ⁻* cells, on the other hand, resembled that of wild-type cells in both resting and glucose stimulating conditions (Fig. 4K). Moreover, in MIN6-K8 WT cells labeled with 10 μg/ml puromycin (Schmidt et al, 2009) global translation, as expected, was increased upon stimulation with high glucose compared to resting conditions, while in *G3BP1⁻ᐟ⁻* cells global translation was higher than in WT cells in both resting and high glucose conditions (Fig. 4L,M).

## Incretin and palmitate resolve G3BP1⁺ condensates

Numerous stimuli foster insulin secretion and/or synthesis (Nauck et al, 2021). Among them are the GLP-1 analogue Exendin-4 (Göke et al, 1993; Alarcon et al, 2006; Gandasi et al, 2018) and palmitate (Usui et al, 2019). Accordingly, treatments with either 50 nM Exendin-4 or 50 μM palmitate reduced the number and area of G3BP1⁺/eIF3b⁺/*Ins1/2* mRNA⁺ condensates in resting MIN6-K8 cells, though not to

the extent observed upon stimulation with high glucose alone (Fig. 5A–C). On the contrary, more, but smaller and with no changes in their area G3BP1⁺/eIF3b⁺/*Ins1/2* mRNA⁺ condensates were observed in cells incubated with 55 mM KCl (Fig. 5A–D), which triggers the secretion of insulin without enhancing its synthesis (Hatlapatka et al, 2009). Glucokinase, or hexokinase IV, catalyzes the conversion of glucose into glucose-6-phosphate in pancreatic beta cells, thus it serves as a glucose sensor for insulin secretion due to its high flux control coefficient on glucose metabolism (Thilagavathi et al, 2022). In resting MIN6-K8 cells treated with 3 μM glucokinase activator Ro-28-1675 G3BP1⁺/eIF3b⁺/*Ins1/2* mRNA⁺ condensates were completely absent, unlike in cells exposed to DMSO alone, indicating that activation of glycolysis, even in the presence of low glucose levels, triggers the resolution of G3BP1⁺/eIF3b⁺/*Ins1/2* mRNA⁺ condensates (Fig. 5A–C).

## G3BP1⁺/ and G3BP2⁺/*Ins1/2* mRNA⁺ condensates are present in resting mouse primary beta cells

To verify the physiological relevance of our findings in MIN6-K8 cells, we investigated the occurrence of G3BP1/2⁺ condensates in primary beta cells and their relationship with *Ins1/2* mRNA. In dispersed mouse pancreatic islet cells exposed to 2.5 or 3.3 mM glucose, both G3BP1⁺ and G3BP2⁺ condensates were present in beta cells, as identified by co-immunostaining for insulin, and enriched in *Ins1/2* mRNA (Fig. 6A,B). Conversely, these condensates were absent in dispersed islet cells incubated with 16.7 mM glucose, in which the signals for both proteins and *Ins1/2* mRNA were more diffused in the cytoplasm (Fig. 6A,B). To preclude the possibility of artifacts arising from the dispersion of pancreatic islets into single cells, similar analyses were conducted on whole mount mouse pancreatic islets. Consistent with the observations made in dispersed islet cells, resting beta cells within isolated islets also displayed G3BP1⁺/*Ins1/2*⁺ mRNA condensates (Fig. 6C,D). Therefore, the phenotype observed within the mouse insulinoma MIN6-K8 cells was validated in primary mouse beta cells.

## *INS* mRNA⁺ condensates are present in pancreatic islet beta cells in situ of non-diabetic living donors

Transcriptomic profiling of laser capture microdissected islets from metabolically phenotyped human living donors in our LIDOPACO cohort had revealed that *G3BP1* is downregulated in individuals with type 2 diabetes compared to normoglycemic donors (Wigger et al, 2021). Hence, we investigated whether G3BP1⁺/*INS* mRNA⁺ condensates could also be detected in beta cells within snap-frozen

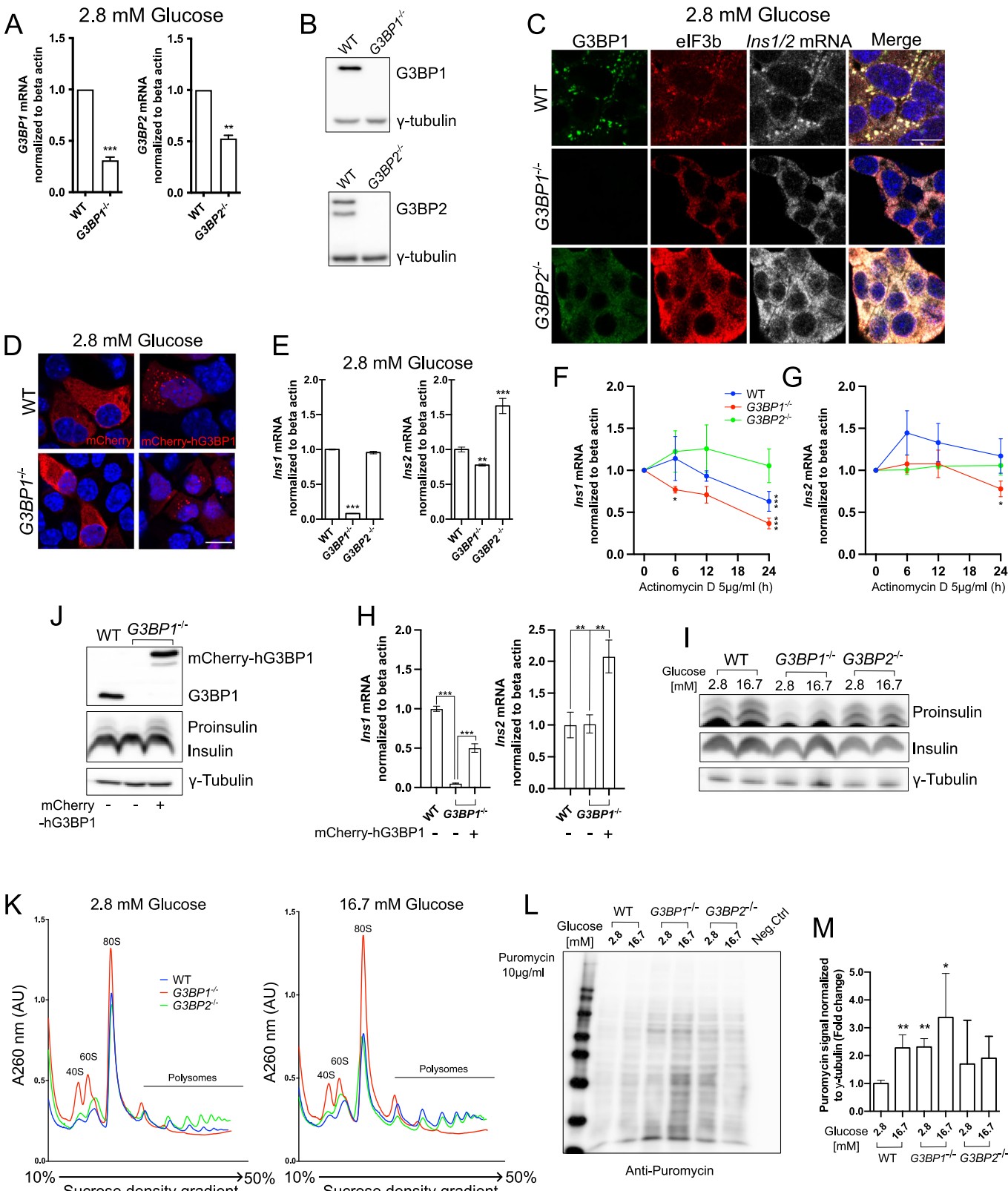

◄ **Figure 4.  G3BP1, unlike G3BP2, safeguards *Ins1/2* mRNA, ensures apt preproinsulin translation, and modulates the polysome turnover in response to glucose variations.**

(A) mRNA levels of *G3BP1* and *G3BP2* in wild type (WT) *G3BP1*$^{-/-}$ and *G3BP2*$^{-/-}$ MIN6-K8 cells as measured by qRT-PCR. (B) Protein levels of G3BP1 and G3BP2 and gamma-tubulin in WT and *G3BP1*$^{-/-}$ and *G3BP2*$^{-/-}$ MIN6-K8 cells as measured by western blot. (C) Immunostaining for G3BP1 (green), eIF3b (red) and smRNA FISH for *Ins1/2* in WT, *G3BP1*$^{-/-}$ and *G3BP2*$^{-/-}$ MIN6-K8 cells at resting glucose concentrations. Nuclei are stained with DAPI (blue). Scale bars = 10 μm (D) Confocal microscopy of *G3BP1*$^{-/-}$ MIN6-K8 cells transiently transfected with *mCherry* or *mCherry-hG3BP1* and exposed to resting glucose conditions. Signals for mCherry or mCherry-hG3BP1 are in red, as labeled in the image. Nuclei are stained with DAPI (blue). (E) Quantification of *Ins1* and *Ins2* mRNA in WT, *G3BP1*$^{-/-}$ and *G3BP2*$^{-/-}$ MIN6-K8 cells in resting glucose conditions as assessed by qRT-PCR. (F, G) Ct values of *Ins1*, *Ins2*, and *Actb* mRNAs in WT, *G3BP1*$^{-/-}$ and *G3BP2*$^{-/-}$ MIN6-K8 cells treated with 5 μg/ml Actinomycin D for 0, 6, 12, and 24 h, as measured by qPCR. (H) Quantification of *Ins1* and *Ins2* mRNA in WT and *G3BP1*$^{-/-}$MIN6-K8 cells transfected with *mCherry* or *mCherry-hG3BP1*. (I) Western blotting for proinsulin, insulin, and gamma-tubulin in WT, *G3BP1*$^{-/-}$ and *G3BP2*$^{-/-}$ in glucose resting or stimulated MIN6-K8 cells. (J) Western blotting for G3BP1, proinsulin, insulin, and gamma-tubulin in WT and *G3BP1*$^{-/-}$ cells transfected with *mCherry* or *mCherry-hG3BP1*. (K) Polysomal profiling of resting or glucose-stimulated WT, *G3BP1*$^{-/-}$ and *G3BP2*$^{-/-}$ MIN6-K8 cells using a 10–50% sucrose gradient to separate and enrich in fractions of the small ribosomal units (40S, first peak), the big ribosomal units (60S, second peak), the monosomes (80S, third peak) and polysome subpopulations starting with 2 ribosomes per polysome (fourth peak) until 7 ribosomes per polysome (last peak). (L, M) Western blotting for global translation assessment with puromycin 10 μg/ml treatment using anti-puromycin antibody in WT, *G3BP1*$^{-/-}$ and *G3BP2*$^{-/-}$ in glucose resting or stimulated MIN6-K8 cells and its quantification. Data information: The intensity of the bands in the Western blots (B) was measured in arbitrary units using ImageStudioLite software, normalized to the γ-tubulin loading control, and fold change was calculated relative to the resting glucose condition. Presented values denote the mean ± SD derived from three independent experiments (except for (D) and (J) which are from one independent experiment), analyzed via one-way ANOVA. Values with *$p < 0.05$, **$p < 0.01$, and ***$p < 0.001$ were considered statistically significant relative to the conditions as shown in each graph. The results of the qRT-PCR are from 3 technical replicates, while for the western blot and the polysome profiling are from 1 technical replicate of each condition per independent experiment. Scale bar = 10 μm. (A): *G3BP1* graph WT vs *G3BP1*$^{-/-}$ ($p = 0.0001$) *G3BP2* graph WT vs *G3BP2*$^{-/-}$ ($p = 0.0037$). (E): *Ins1* graph WT vs *G3BP1*$^{-/-}$ ($p < 0.0001$). *Ins2* graph WT vs *G3BP1*$^{-/-}$ ($p = 0.004$); *G3BP2*$^{-/-}$ ($p < 0.0001$). (F): *Ins1* graph WT 0 h vs WT 24 h ($p < 0.0001$), *G3BP1*$^{-/-}$ 0 h vs *G3BP1*$^{-/-}$ 6 h ($p = 0.0363$); 24 h ($p < 0.0001$). *Ins2* graph *G3BP1*$^{-/-}$ 0 h vs *G3BP1*$^{-/-}$ 24 h ($p = 0.0159$). (H): *Ins1* graph both comparison $p < 0.0001$. *Ins2* graph WT vs *G3BP1*$^{-/-}$ mCherry-hG3BP1 ($p = 0.0013$), *G3BP1*$^{-/-}$ vs *G3BP1*$^{-/-}$ mCherry-hG3BP1 ($p = 0.0014$). (M): 2.8 WT vs 16.7 WT ($p = 0.0058$); 2.8 *G3BP1*$^{-/-}$ ($p = 0.0051$); 16.7 *G3BP1*$^{-/-}$ ($p = 0.0215$). Source data are available online for this figure.

surgical specimens of living donors undergoing pancreatectomy for a variety of disorders of the exocrine pancreas. Initial attempts to identify such condensates through a random survey of samples from normoglycemic living donors (NLD) were unfruitful. This was not unexpected, as the threshold for stimulation of insulin translation and secretion of human beta cells is ~4 mM glucose, i.e., lower than in rodents, while even at fast the average glycemia of normoglycemic subjects in our cohort (65) was 5.03 ± 0.32 mM. For this reason, we took advantage of the continuous monitoring of glycemia during surgery to select among the NLD within our cohort those with the lowest intraoperative glycemia in the minutes before pancreas resectomy. Intriguingly, in specimens of two such donors, coded NLD1 and NLD2, several pancreatic islets displayed *INS* mRNA$^+$ condensates, which especially in the case of NLD1, partially co-localized with G3BP1 (Fig. 7). This pattern, however, was not observed in specimens of other non-diabetic donors with higher intraoperative glycemia (Fig. 7). Nonetheless, these findings document the occurrence of G3BP1$^+$/*INS* mRNA$^+$ condensates in human beta cells in situ, pointing to the physiological significance of such dynamic compartments for control of insulin translation.

## Discussion

In this study, we unveil the presence of G3BP1/2$^+$ and eIF3b$^+$ cytosolic condensates, which encapsulate *Ins1/2* mRNAs at resting glucose concentrations and dissolve when beta cells are exposed to higher glucose levels. Intriguingly, none of the mRNAs encoding for other glucose-induced insulin secretory cargoes *Ptprn*, *Pcsk1*, *Pcsk2* were found within G3BP1$^+$ condensates, suggesting the potentially selective confinement of *Ins1/2* mRNA in such novel beta cell compartment. As stress granules can capture various mRNAs, including housekeeping ones like *GAPDH* mRNA (Kedersha and Anderson, 2002), the condensates described here may not be equivalent to conventional stress granules induced by

stress conditions. However, *Ins1/2* transcripts are by far the most abundant in beta cells, which might account for their accumulation and detectability into G3BP1$^+$ condensates.

By synthesizing and secreting insulin when exposed to 16.7 mM glucose, but not when resting with 2.8 mM glucose (Iwasaki et al, 2010), MIN6-K8 cells are a well suited model to study the dynamics of G3BP1$^+$ condensates. Due to their occurrence in the presence of 2.8 mM glucose, G3BP1$^+$ condensates cannot be readily attributed to stress resulting from glucose deprivation. However, since MIN6-K8 cells are cultured in media with 25 mM glucose, the drastic change to resting glucose levels could represent a potential stress. Therefore, we tested the effect of sodium arsenate, a known inducer of stress granules. In its presence, G3BP1$^+$ condensates persisted even in the presence of 16.7 mM glucose. Notably, G3BP1$^+$ condensates resolved rapidly, i.e., within 30 min, following glucose stimulation, in contrast to canonical stress granules, which in HeLa cells require >2 h to disappear upon removal of sodium arsenate (Tolay and Buchberger, 2021). Hence, beta cells may use a similar machinery to stress granules for physiological purposes.

Stress granules may mainly form as a consequence of polysome disassembly, which can be elicited in several ways: phosphorylation of eIF2α, inhibition of mTOR or interference with the eIF4F complex (Hofmann et al, 2021). mTOR inhibition can be promoted by phosphorylation of AMPKα on Thr172 by AXIN/LKB1 (Lin and Hardie, 2018). This event occurs in low glucose concentrations when the glycolytic enzyme aldolase, unbound to its substrate FBP, recruits AXIN to v-ATPase, Ragulator, and AMPKα, forming a supramolecular complex on lysosome surfaces, leading to LKB1-mediated AMPKα phosphorylation. Concurrently, AMPKα is phosphorylated through the canonical pathway due to an increase in the AMP/ADP ratio resulting from low glucose, which in turn decreases glycolytic flux. We observed that in low glucose conditions, G3BP1$^+$ condensates appear and AMPKα phosphorylation is elevated in comparison to cells exposed to high glucose levels, consistent with findings that AMPKα levels are rapidly

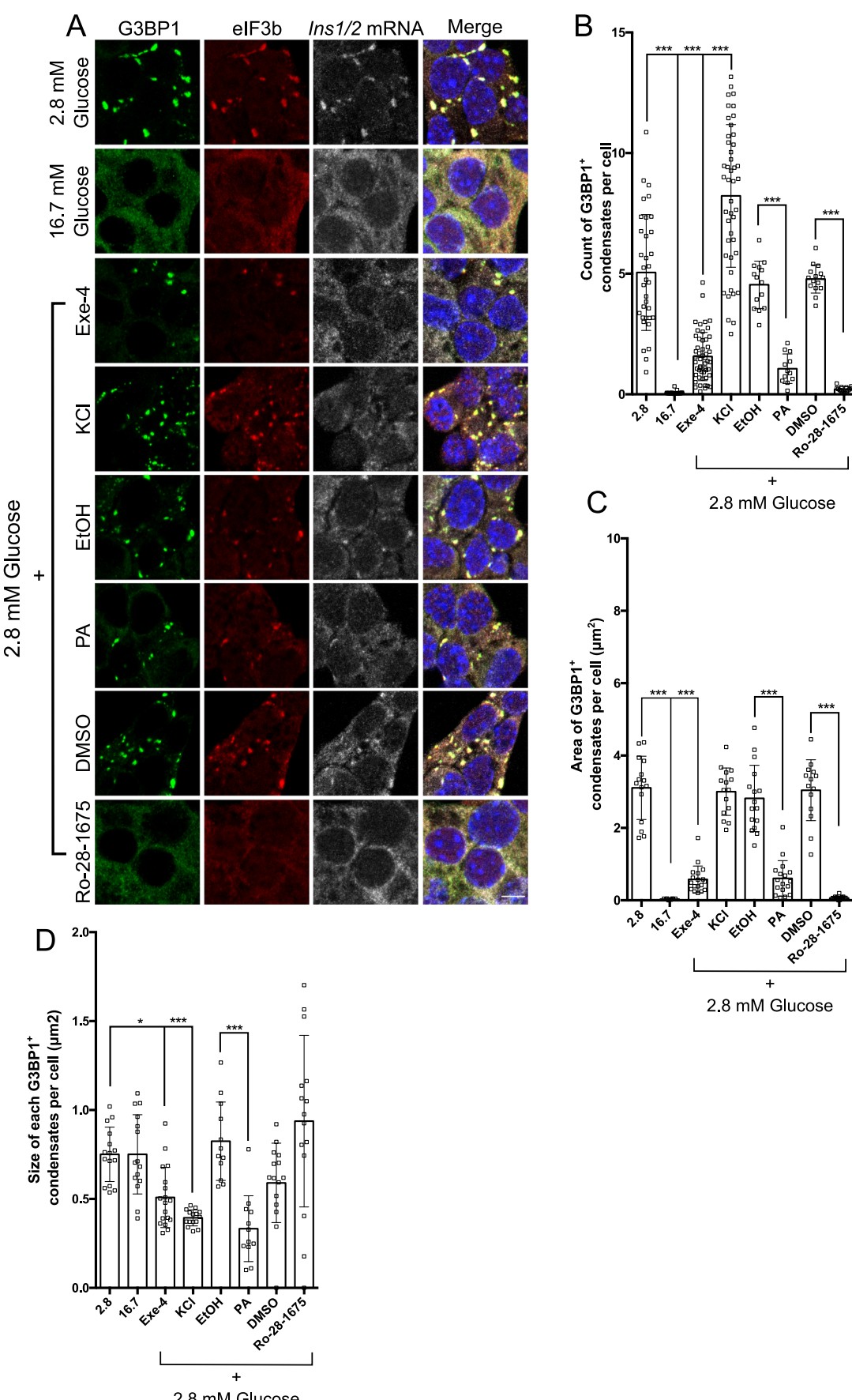

**Figure 5.    Impact of various insulin secretagogues on G3BP1⁺ condensates.**

(A) Immunostaining for G3BP1 (green), eIF3b (red), and smRNA FISH for *Ins1/2* (gray) in MIN6-K8 wild-type cells treated with Exendin-4 (50 nM), palmitate (50 μM), KCl (55 mM) and glucokinase activator Ro-28-1675 (3 μM) in resting glucose conditions. (B) Quantification of the count of G3BP1⁺ condensates per cell in MIN6-K8 cells in resting glucose concentrations upon addition of Exendin-4 (50 nM), palmitate (50 mM), KCl (55 mM) or Ro-28-1675 (3 μM). (C) Area of G3BP1⁺ condensates per cell (μm²) in all tested conditions as in (B). (D) Size of each G3BP1⁺ condensate (μm²) in all tested conditions as in (B). Data information: Presented values denote the mean ± SD derived from three independent experiments, analyzed via one-way ANOVA. Values with $*p < 0.05$, $**p < 0.01$, and $***p < 0.001$ were considered statistically significant relative to the conditions as shown in each graph. Quantifications from (B–D) were taken from ≥15 images (at least 5 per independent experiment). Scale bar = 5 μm. (B, C): Every comparison $p < 0.0001$. (D): 2.8 vs Exe-4 ($p = 0.0297$); KCl ($p = 0.0007$), EtOH vs PA ($p = 0.0001$). Source data are available online for this figure.

reduced, plausibly due to degradation upon glucose stimulation. Furthermore, a non-canonical pathway of stress granule formation involving AMPKα phosphorylation induced by energy depletion has been discovered in HeLa cells, where phospho-AMPKα^Thr172 interacts directly with G3BP1 (Mahboubi and Stochaj, 2017). We were able to detect phospho-AMPKα^Thr172 signal in G3BP1⁺ condensates in resting glucose (Fig. 2G), therefore resembling the described mechanism. In addition, aldolase inhibition with the specific inhibitor aldometanib which resembles aldolase not being bound to its substrate FBP (Zhang et al, 2022) led to an increase in G3BP1⁺ condensates presence even in high glucose of 16.7 mM and with the addition of pyruvate 20 mM, conditions in which in the absence of aldometanib G3BP1⁺ condensates are dissolved (Fig. 3B). Based on this literature and our findings, there seems to be a connection in insulinoma MIN6-K8 cells between low glucose, AMPKα-phosphorylation/activation, and presence of G3BP1⁺ condensates, via the aldolase-phospho-AMPKα^Thr172-G3BP1 axis. Evidence that inhibition of aldolase prevents the resolution of G3BP1⁺/*Ins1/2*⁺ mRNA condensates and the elevation of ATP levels, even upon the exogenous supply of pyruvate, indicates that beta cells strictly couple preproinsulin translation to the rate of glycolysis, which is enhanced upon glucose stimulation. Aldolase acts therefore as a gatekeeper in this process by exerting a control on mitochondria (Fig. 8), conceivably through the NADH produced from metabolism of glyceraldehyde-3 phosphate downstream of aldolase. NADH is indeed a significant contributor to the mitochondrial pool of reducing equivalents required for the proper function of the electron transport chain, and thus for mitochondrial oxidative phosphorylation and ATP production. Such mechanism may serve the need of beta cells to exquisitely upregulate insulin production and secretion, which depends on the increase of the ATP/ADP ratio (Ashcroft, 2023), in response to hyperglycemia only and not to the elevation of other fuels in order to prevent hypoglycemia, which can be fatal. Beta-oxidation of fatty acids, in particular, elevates ATP levels, but does not result in the generation of cytosolic NADH. Notably, we discovered AldoB, which is physiologically barely expressed in adult, mature beta cells, to be among the most upregulated genes and proteins in islets of subjects with type 2 diabetes, with its levels being tightly correlated with those of glycosylated hemoglobin (HbA1c), the most reliable biomarker of hyperglycemia (Gerst et al, 2018; Solimena et al, 2018; Wigger et al, 2021). Therefore, it will be important to investigate further how AldoB overexpression in islets of patients with type 2 diabetes affects beta cell AMPKα/mTOR activities, mitochondrial function, the dynamics of G3BP1⁺/*INS* mRNA condensates, and thus insulin production and secretion.

Despite both *G3BP1*⁻/⁻ and *G3BP2*⁻/⁻ MIN6-K8 cells lack the ability to form condensates, we found deletion of *G3BP1* or *G3BP2*

to have contrasting effects on *Ins1/2* transcript variants and proinsulin levels when compared to wild-type cells. In *G3BP1*⁻/⁻ cells, unlike in *G3BP2*⁻/⁻ cells, both *Ins* transcripts and proinsulin levels were decreased. Rasputin, the Drosophila homolog of G3BP1, has been shown to bind and stabilize short mRNAs implicated in transcription, splicing, and translation during embryogenesis, thereby enhancing their translation (Laver et al, 2020). Additionally, expression of a dominant-negative variant of G3BP1 lacking the NTF2L domain, and consequently unable to dimerize or form condensates, activated intra-axonal mRNA translation and axon growth in rat dorsal root ganglia (DRG) neurons, hence expediting nerve regeneration in vivo (Sahoo et al, 2018). This data underscores a role for G3BP1 beyond stress responses, highlighting its function in physiological conditions where it may bolster mRNA stability and translation. Correspondingly, reduction of *Ins1/2* transcript and proinsulin levels in *G3BP1*⁻/⁻ cells only indicates that G3BP1, but not its close paralogue G3BP2, play a role in stabilizing and enhancing the translation of *insulin* mRNAs. Moreover, impaired glucose stimulated insulin secretion in *G3BP1*⁻/⁻ cells is concordant with evidence of *G3BP1* downregulation in islets of patients with type 2 diabetes, who suffer from reduced insulin release. The transient expression of an mCherry-hG3BP1 construct effectively rescued the phenotype observed in *G3BP1*⁻/⁻ cells, including the presence of mCherry-hG3BP1 in resting glucose conditions and the restoration of proinsulin content to levels comparable to those detected in wild-type MIN6-K8 cells. Additionally, *Ins1* and *Ins2* mRNA expression were recovered, with their levels being approximately 50 and 200% of those measured in wild-type MIN6-K8 cells, respectively.

Previous studies suggested that both *G3BP1* and *G3BP2* are required for stress granule assembly (Kedersha et al, 2016; Yang et al, 2020). We found that deletion of either *G3BP1* or *G3BP2* alone in MIN6-K8 cells was sufficient to elicit this phenotype. *G3BP1*⁻/⁻ MIN6-K8 cells carry a 25 bp deletion in *G3BP1* exon 2, potentially resulting in the expression of a truncated mutant encompassing 56 residues in total, 23 of which belong to the G3BP1 NTF2L domain (Fig. EV6A). Whether such putative peptide may still dimerize with G3BP2 and thereby inhibit stress granule formation cannot be excluded. Notably, it was shown that truncation of USP10, a G3BP1 binding protein which is also essential for stress granule formation (Takahashi et al, 2013), generated a peptide which precludes the occurrence of stress granules (Kedersha et al, 2016). In the case of *G3BP2*⁻/⁻ MIN6-K8 cells, the 12 bp deletion in exon 2 results in the removal of residues 36–39 (Fig. EV6B), which according to AlphaFold Structure Prediction, lie in the first beta sheet of the protein, likely affecting its folding and stability. We remark that in the aforementioned studies stress granules were induced by pharmacological agents

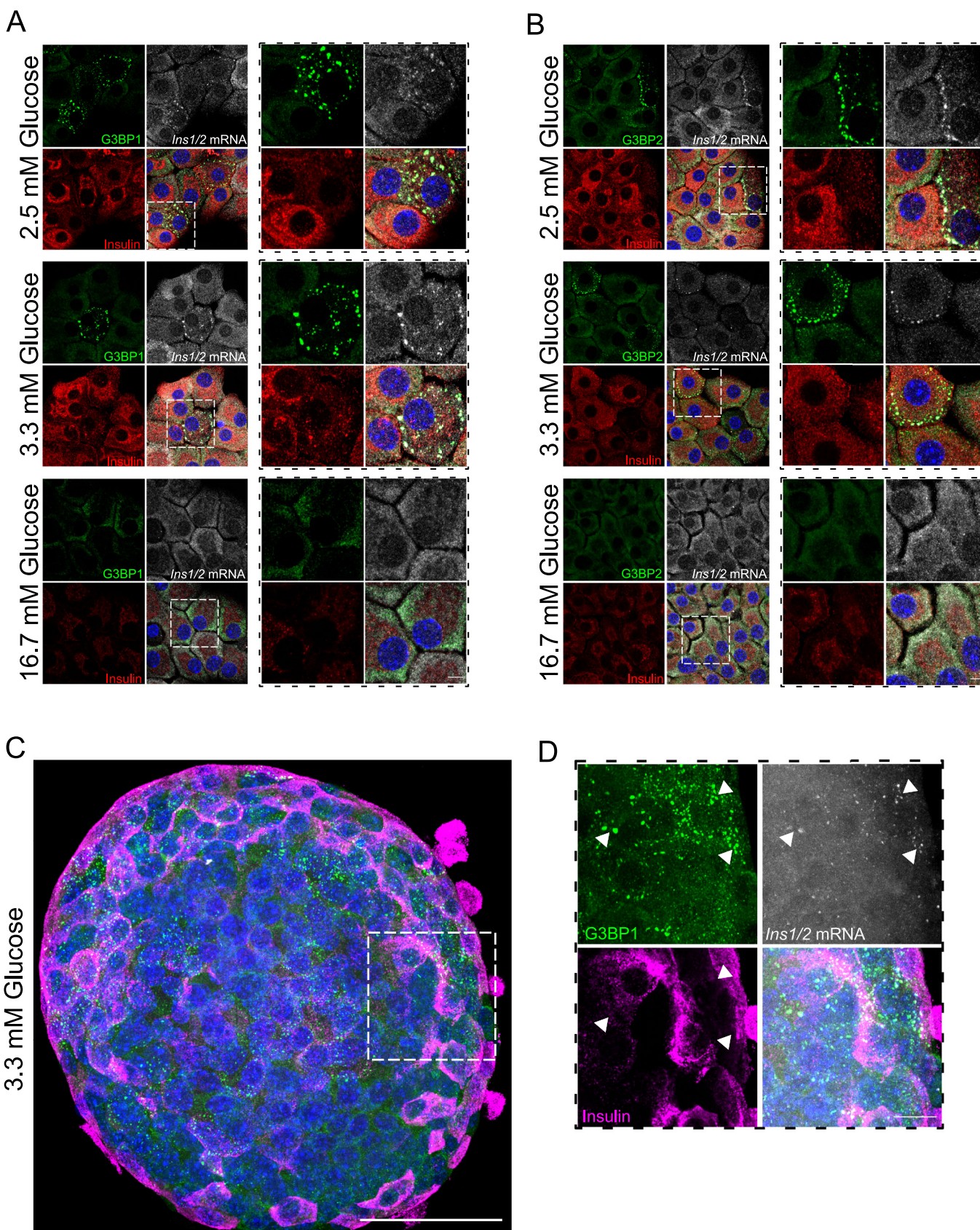

**Figure 6.   Detection of G3BP1⁺, G3BP2⁺ and *Ins1/2* in dispersed and whole mouse pancreatic islet cells under resting glucose conditions.**

(A, B) Immunostaining for G3BP1 (green, left panel), G3BP2 (green, right panel), Insulin (red), and smRNA FISH for *Ins1/2* (gray) in dispersed mouse pancreatic islet cells under glucose resting (2.5 and 3.3 mM) or stimulating (16.7 mM) conditions. Nuclei are stained with DAPI (blue). (C, D) Immunostaining for G3BP1 (green), Insulin (magenta) and smRNA FISH for *Ins1/2* (gray) in whole mount mouse pancreatic islets under resting conditions. Arrowheads point to G3BP1⁺/*Ins1/2* mRNA⁺ condensates. Nuclei are stained with DAPI (blue). Data information: The data presented was derived from three independent experiments. Scale bars in (A), (B) and (D) = 5 μm; in (C) = 50 μm. Source data are available online for this figure.

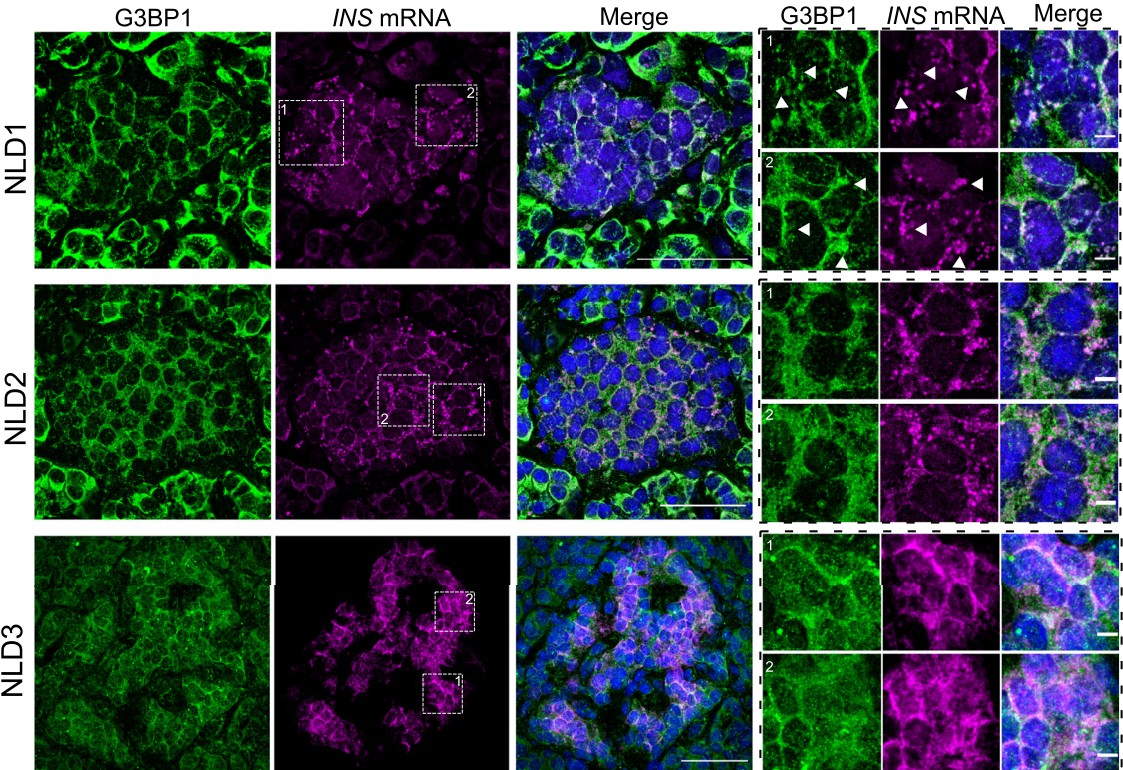

**Figure 7.   G3BP1⁺ and *INS* mRNA⁺ condensates in beta cells in situ of normoglycemic living donors.**

Immunostaining for G3BP1 (green), and smRNA FISH for *INS* (magenta) in human pancreatic cryosections from normoglycemic living donors (NLD) 1, 2 and 3. Nuclei are stained with DAPI (blue). For better visualization, the right panels show magnified crops of the area selected with white line squares in the left panels. Arrowheads point to G3BP1⁺/*INS* mRNA⁺ condensates. Data information: Scale bars = 50 μm or 5 μm in the left and right panels, respectively. Source data are available online for this figure.

triggering eIF2α phosphorylation (sodium arsenate, clotrimazole, thapsigargin) or eIF4A inactivation (pateamine-A, rocaglamide A), and not by modulating glucose concentrations within a range mimicking physiological condition.

G3BP1 has also been identified as a key player in the ribosome-associated quality control, overseeing the fidelity of mRNA translation, particularly when aberrant translation occurs (Meyer et al, 2020). Beta cells have an outstanding translation performance, as they can translate >3 × 10³ preproinsulin molecules/second/beta cell (Schuit et al, 1991; Vasiljević et al, 2020), hence necessitating a highly efficient surveillance mechanism. Translation can be compromised by degradation via non-stop decay (NSD) (Van Hoof et al, 2002; Vasudevan et al, 2002; Meyer et al, 2020), or in no-go decay (NGD) (Harigaya and Parker 2010; Shoemaker and Green 2012; Meyer et al, 2020). Both pathways are triggered by collisions of ribosomes, prompting their disassembly (Simms et al, 2017; Juszkiewicz et al, 2018; Meyer et al, 2020). In such instances, G3BP1

forms complexes with the G3BP1-family member USP10 to deubiquitinate 40S ribosomal proteins, safeguarding them from lysosomal degradation and maintaining ribosomal subunit balance (Meyer et al, 2020). In the absence of the protective deubiquitination activity of G3BP1-family-USP10 complexes, the 40S proteins along with *Ins1/2* transcripts may be degraded. Furthermore, as preproinsulin biosynthesis occurs in polysomes (Wolin and Walter, 1993) the lack of a dynamic polysomal response in glucose-stimulated G3BP1⁻/⁻ MIN6-K8 cells is consistent with their reduced proinsulin translation and content.

Glucose is key for insulin secretion and biosynthesis (Leibiger et al, 2000). Yet, other molecules, such as the incretin hormone glucagon-like peptide-1 (GLP-1), potentiates glucose-stimulated insulin production and release (Thorens, 1992; Alarcon et al, 2006; Nauck et al, 2021). Accordingly, exposure of resting MIN6-K8 cells to the GLP1-analogue Exendin-4 reduced the number and size of G3BP1⁺ condensates, pointing to a priming effect of GLP-1 on beta

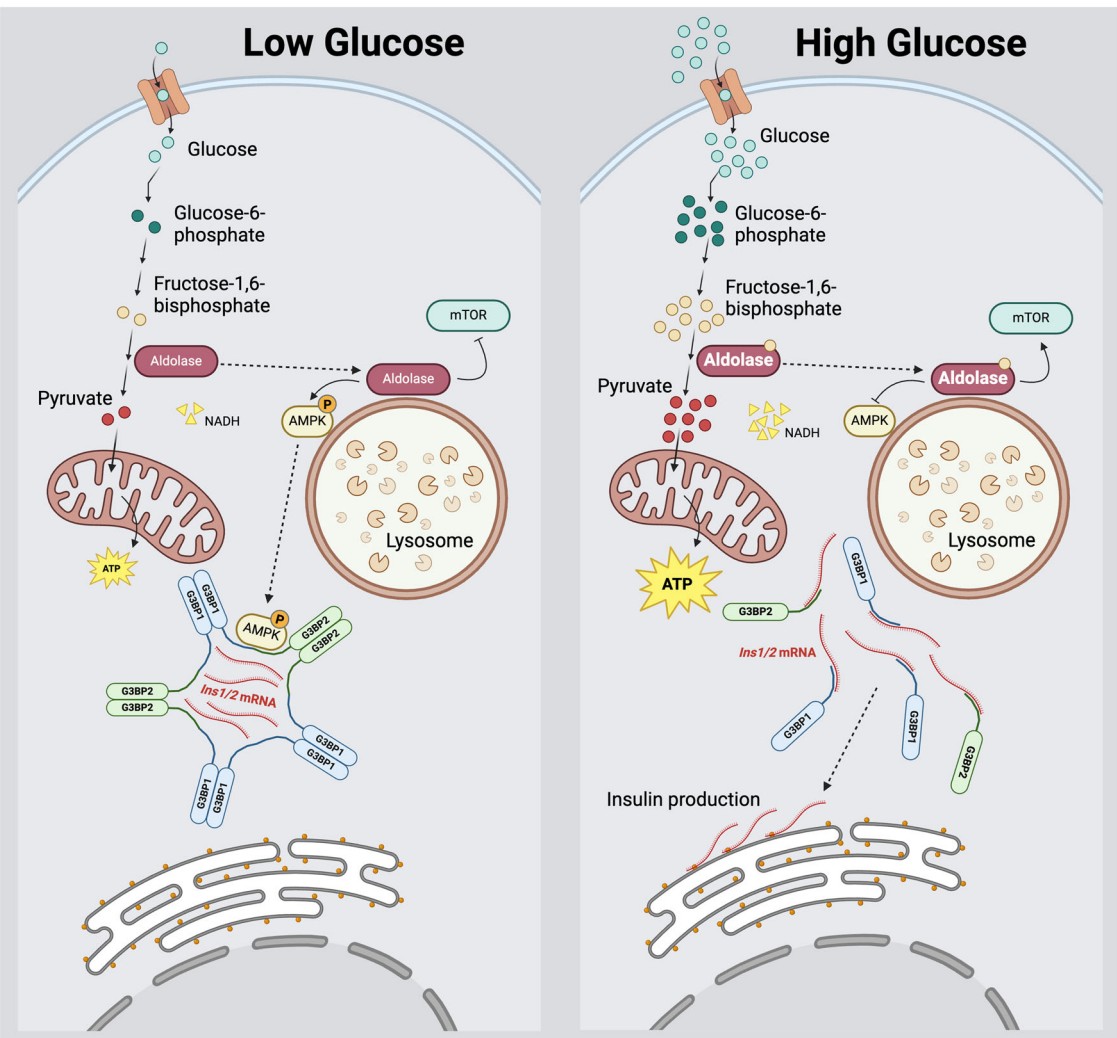

**Figure 8. Aldolase activity regulates G3BP1/2⁺ condensates in beta cells.**

In resting beta cells (left panel), aldolase activity, NADH levels, oxidative phosphorylation and ATP production are modest, while pAMPKα is elevated and found on G3BP1/2⁺ condensates. Accordingly, the rate of insulin translation is low. Glucose stimulation (right panel) enhances aldolase activity, NADH levels and thereby oxidative phosphorylation and ATP production, while reducing the levels of pAMPKα and resolving G3BP1/2⁺ condensates, hence fostering the translation of insulin mRNA.

cell responsiveness to elevation of glycemia (Göke et al, 1993). This finding is particularly interesting, considering previous literature stating that increased glycolytic flux, where aldolase is substrate-bound, inactivates AMPKα (Lin and Hardie, 2018). Free fatty acids (FFAs), such as palmitate, induce insulin secretion from isolated human pancreatic islets exposed to physiological fasting glucose levels, a process linked to increased glycolytic flux and mitochondrial respiration (Cen et al, 2016). This mechanism might explain the observed reduction in G3BP1⁺ condensates with the addition of palmitate to resting glucose levels. Similar considerations could apply to the reduction of G3BP1⁺ condensates upon treatment with the allosteric glucokinase activator Ro-28-1675 (Grimsby et al, 2003), which enhances glycolysis. Exposure of beta cells to high levels of KCl is commonly used to induce membrane depolarization and stimulate insulin secretion (Brüning et al, 2017). Unlike glucose, however, this treatment does not promote insulin production (Hatlapatka et al, 2009). Accordingly, KCl treatment

did not induce the loss of G3BP1⁺ condensates but rather increased their number. Sulfonylureas, the oldest class of medications for the treatment of type 2 diabetes, akin to KCl treatment, induce membrane depolarization and insulin secretion by binding to the sulfonylurea receptor subunit on ATP-sensitive potassium channels (Costello et al, 2024). Despite their insulin secretagogue activity, these drugs are now considered secondary line treatments due to their long-term detrimental effects on beta cell viability, as they stimulate the release of insulin, but not its biosynthesis (Mohan et al, 2022). Furthermore, sulfonylureas have been linked to endoplasmic reticulum stress (Qian et al, 2008), and increased ROS production, leading to oxidative stress (Tsubouchi et al, 2004). Consequently, the higher prevalence of G3BP1⁺ condensates in KCl-treated cells compared to treatment with resting glucose alone could be indicative of a stress response.

Having access to pancreatic tissue from a large cohort of normoglycemic living donors who underwent pancreatic surgery

enabled us to detect cytosolic *INS* mRNA$^+$ granular structures that partially co-localized with G3BP1$^+$ condensate-like compartments, albeit not as distinctly as seen in mouse beta cells. This observation is particularly noteworthy considering the complexity of the physiological and clinical context. Although fasted preoperatively, these patients may indeed receive various treatments during the long surgical procedure of pancreatectomy, including the infusion of glucose (Table EV3) and insulin, hence creating conditions that significantly differ from the controlled environments of our experiments in vitro.

# Methods

### Reagents and tools table

| Reagent/Resource | Reference or Source | Identifier or Catalog Number |
|---|---|---|
| **Experimental models** | | |
| MIN6 K8 cells | Susumo Seino lab | Iwasaki et al, 2010 |
| Mouse C57JBL/6J strain | | |
| Human pancreatic islets | University Hospital Dresden | |
| **Recombinant DNA** | | |
| Specific qPCR primers | Eurofins | Table EV-2 |
| pSpCas9(BB)-2A-GFP (PX458) | Addgene | Cat # 48138 |
| pmCherry-N1 | Clontech | Cat # 632523 |
| pcDNA3.1-mCherry-hG3BP1 | Simon Alberti lab | Trussina et al, 2024 |
| **Antibodies** | | |
| Specific primary antibodies | | Table EV-1 |
| Specific secondary antibodies | | Table EV-1 |
| **Oligonucleotides and other sequence-based reagents** | | |
| Probes for *Ins1/2* transcripts | Biosearch Technologies | Cat # VSMF-3612-5 |
| Probes for *LacZ* | Biosearch Technologies | Cat # VSMF-1002-5 |
| Probes for *Ptprn*, *Pcsk1* and *Pcsk2* | Biosearch Technologies | Table EV-4 |
| **Chemicals, Enzymes and other reagents** | | |
| Sodium hydrogen arsenate heptahydrate | Alfa Aesar | Cat # 33373.14 |
| Palmitic acid | Sigma | Cat # P5585 |
| Exendin-4 | Sigma | Cat # E7144 |
| Ro-28-1675 | MedChemExpress | Cat # HY-10595 |
| Puromycin | InvivoGen | Cat # ant-pr-1 |
| Actinomycin D | Sigma-Aldrich | Cat # SBR00013 |
| Lambda Protein Phosphatase | NEB | Cat # P0753L |
| M-MLV Reverse Transcriptase | PROMEGA | Cat # M1705 |
| DAPI | Sigma | Cat # D9564 |
| Mowiol | Calbiochem | Cat # 475904 |

| Reagent/Resource | Reference or Source | Identifier or Catalog Number |
|---|---|---|
| Aldometanib | MedChemExpress | Cat # LXY-05-029 |
| Dako antibody diluent | DAKO | Cat # S0809 |
| Rotenone | Sigma | Cat # R8875 |
| **Software** | | |
| Fiji software | https://imagej.net/software/fiji/ | |
| GraphPad Prism | https://www.graphpad.com/features | |
| **Other** | | |
| Pierce™ BCA Protein Assay Kit | ThermoFisher | Cat # 23225 |
| SuperSignal™ West Pico PLUS Chemiluminescent Substrate | ThermoFisher | Cat # 34580 |
| RNeasy kit | Qiagen | Cat # 74104 |
| GoTaq. qPCR Mastermix | PROMEGA | Cat # A6002 |
| Insulin Ultra-Sensitive Assay kit | Cisbio | Cat # 62IN2PEG |
| Rat/Mouse Proinsulin ELISA | Mercordia | Cat # 10-1232-01 |
| CellTiter-Glo. 2.0 Cell Viability Assay | Promega | Cat # G9241 |
| Multiplex Fluorescent Reagent Kit v2 | ACD | Cat # 323100 |
| Hybridization buffer | Biotechne, | Cat # SMF-HB1 |
| BD FACSArialII | BD Bioscience | |
| AriaMx RealTime PCR System | Agilent Technologies | |
| Amersham Imager 600 | GE Healthcare | |
| Grandient Master 108 | Biocomp | |
| Piston Gradient Fractionator | Biocomp | |
| Sirius Luminometer | Berthold Detection Systems | |
| Synergy Neo2 multi-mode reader | BioTek | |
| Nikon C2+ Confocal | Nikon | |
| LSM980 Airy Scan | Zeiss | |

## Mice

Animal husbandry was done in accordance with applicable animal welfare legislation (e.g., RL 2010/63/EU). Experiments were licensed by the respective authorities in Sachsen (protocols TV A 6/2018 and TV vG 10/2023). Mice were housed in individually ventilated cages in a 12 h light and 12 h dark cycle with food and water ad libitum.

## Human living donors and pancreatic tissue

Pancreatic surgical tissue from the "healthy" resection margins was obtained from living donors who underwent pancreatectomy at our

University Hospital due to various disorders of the exocrine pancreas. All donors provided their informed consent for the use of surgical and blood samples for research purposes and our studies were conducted with the ethical approval of the Ethical Committee of the Technische Universität Dresden under the license EK151062008. Donors underwent an oral glucose tolerance test in the days immediately preceding pancreatectomy and their glycemia was monitored during surgery. After excision, the pancreatic tissue was immediately snap-frozen, then embedded in OCT and stored at −80 °C. Cryosections of 10 µm were generated with a cryostat (Epredia™ CryoStar™ NX50 Kryostat, Fisher Scientific) for further analysis.

## Culture of MIN6-K8 cells

MIN6-K8 insulinoma cells (Iwasaki et al, 2010) were cultured in DMEM High-Glucose (25 mM) medium (ThermoFisher, 11965092) supplemented with 15% fetal bovine serum (FBS) (Gibco, A5258701), 1% penicillin-streptomycin and 70 mM 2-Mercaptoethanol at 37 °C in a humidified atmosphere with 5% $CO_2$. The culture media was sterilized using a Rapid Flow Filter Unit (ThermoFisher, 566-0020). For cell expansion, 175 cm² flask bottles (Corning, 353112) were used, and cells were subsequently seeded in various plates depending on the specific techniques being employed further in the experiments.

## Treatments of MIN6-K8 cells with different glucose buffers

MIN6-K8 insulinoma cells grown in 25 mM glucose were first equilibrated to a resting state with 2.8 mM glucose at 37 °C for 30 min, and then either exposed without washing to a stimulation buffer with 16.7 mM glucose or kept in fresh resting glucose buffer at 37 °C for 30 min. Resting glucose buffer was composed of 15 mM Hepes pH 7.4, 5 mM KCl, 2.8 mM Glucose, 60 mM NaCl, 24 mM $NaHCO_3$, 1 mM $MgCl_2$, 2 mM $CaCl_2$, 1 mg/ml bovine serum albumin (Roth, 8076.3). Stimulation glucose buffer was composed of 15 mM Hepes pH 7.4, 5 mM KCl, 25 mM Glucose, 60 mM NaCl, 24 mM $NaHCO_3$, 1 mM $MgCl_2$, 2 mM $CaCl_2$, 1 mg/ml bovine serum albumin (Roth, 8076.3). Both buffers were sterile filtered using Rapid Flow Filter Unit and kept at 4 °C. Buffers were warmed in a water bath at 37 °C for at least 15 min before performing the treatments. MIN6-K8 cell culture media was washed twice with PBS 1X before adding the resting glucose buffer. After the treatment cells were processed in different ways depending on the desired technique to be performed downstream.

## Treatments of MIN6-K8 cells with different insulin secretagogues

MIN6-K8 cells were treated with various reagents to assess stress responses and activation mechanisms, using the protocols outlined in the methods. Sodium hydrogen arsenate heptahydrate (Alfa Aesar, 33373.14) at a concentration of 1 mM was added to either resting or stimulation glucose buffers to induce oxidative stress. Palmitic acid (Sigma, P-5585) was prepared as a 10 mM stock in absolute ethanol and used at 50 µM, Exendin-4 (Sigma, E7144) was prepared as a 50 mM stock in autoclaved water and used at 50 nM, Ro-28-1675 (MedChemExpress, HY-10595) was kept as a 10 mM stock in DMSO and used at 3 µM, and KCl was used at 55 mM; all these reagents were added exclusively to resting glucose buffer.

Each treatment was conducted in parallel with controls that included the respective vehicle in the resting buffer.

## Puromycin and Actinomycin D treatments

To monitor global translation, MIN6-K8 cells were labeled with 10 µg/ml puromycin (InvivoGen, ant-pr-1) for 10 min when the resting or stimulation treatment times were over (Schmidt et al, 2009). Global translation was assessed by western blotting of puromycin treated and untreated cell extracts with a mouse anti-puromycin antibody (Sigma-Aldrich, ZMS1016). Wild-type, $G3BP1^{-/-}$ and $G3BP2^{-/-}$ MIN6-K8 cells were treated with 5 µg/ml Actinomycin D (Sigma-Aldrich, SBR00013) to measure mRNA degradation. Actinomycin D was added to the cell culture media for different timepoints as shown in the Results section.

## Protein extraction and quantification

MIN6-K8 cells were harvested from 6-well plates at a confluence of 80–90%. Cells were scraped from each well and pelleted in Eppendorf tubes in the centrifuge at 4 °C × 5000 rpm for 2 min in PBS 1X. Then, PBS was removed and 30–50 µL of lysis were added depending on if the pellet was from 1 or 2 wells. For quantifying total protein concentration, the Pierce™ BCA Protein Assay Kit (ThermoFisher, 23225) was used as described by the manufacturer. Samples were kept at −80 °C for long-term storage.

## SDS-PAGE and Western blots

Protein samples (20–30 µg) were subjected to SDS-PAGE (Sodium Dodecyl Sulfate-Polyacrylamide Gel Electrophoresis) for separation based on molecular weight. Proteins were first denatured for 5 min at 95 °C. The samples were then loaded onto a polyacrylamide gel in specific % depending on the target protein and electrophoresis was carried out at a constant 180 volts for 2 h. Following electrophoresis, proteins were transferred from the gel onto a nitrocellulose membrane in a transfer buffer in a wet transfer system for 2 h at 400 mA. After the transfer, membranes were blocked with a solution of 5% fat free dehydrated milk in PBST 0.1%. Subsequently, the membrane was incubated with horseradish peroxidase (HRP)-conjugated secondary antibodies in a dilution of 1:6000 in PBST-T 0.1 and 5% fat free dehydrated milk. Protein bands were visualized by treating the membrane with SuperSignal™ West Pico PLUS Chemiluminescent Substrate (ThermoFisher, 34580) as described in the kit protocol and then images were taking using the Amersham Imager 600. See Table EV1 for further details about gel %, primary antibodies dilution and catalog numbers.

For lambda phosphatase assay the reactive from Lambda Protein Phosphatase (Lambda PP) (NEB, P0753L) were used and followed the protocol from the manufacturer.

## RNA extraction, cDNA preparation and quantitative PCR (qPCR)

MIN6-K8 cells were harvested from 6-well plates at a confluence of 80–90%. Cells were scraped from each well and pelleted in Eppendorf tubes in the centrifuge at 4 °C × 5000 rpm for 2 min in PBS 1x. RNA extraction from cell pellets was performed by using the RNeasy kit (Qiagen, 74104) following the manufacturer's protocol. The

concentration of total RNA (ng/ml) per sample was measured with the Nanodrop TM1000 spectrophotometer from Thermo.

For the synthesis of complementary DNA (cDNA), a total of 1 µg of total RNA was used from each sample. Reactions were prepared using the M-MLV Reverse Transcriptase protocol (Promega, M1705) as described by the manufacturer. Enzyme was added to the reaction 10 min after incubating all the components at 72 °C, to avoid heat inactivation. Then, for the reverse transcription reaction samples were incubated for 40 min at 42 °C and then last incubation was for 10 min at 70 °C. After the reaction was done, 1 volume of water was added to the final reaction (40 µl final volume).

For quantitative PCR reactions (qPCR), reagents from GoTaq® qPCR Mastermix (Promega, A6002) were used. Reactions were prepared in technical duplicates and based on manufacturer's instructions using 1 µl of cDNA for each reaction. qPCR reactions were conducted in 96-well plates in the AriaMx Real-Time PCR System from Agilent. Genes evaluated and their respective primer sequences can be found in Table EV2. Fold changes between conditions were calculated based on the deltadeltaCt method as described by (Livak and Schmittgen, 2001).

## Immunohistochemistry and RNA fluorescent in situ hybridization (RNA-FISH)

For performing immunohistochemistry only, MIN6-K8 cells at 70–80% of confluence grown on 12 mm glass coverslips (Roth, P231.1) in 24-well plates (Corning, 353047) were fixed with 4% PFA (Santa Cruz, sc-281692) for 20 min under the chemical hood. Fixed cells were immediately stained or stored at 4 °C for up to 2 weeks in PBS-Azide 0.02%. Fixed cells were permeabilized with PBS-Triton 0.1% for 15. Then, cells were blocked with blocking solution (0.2% Fish skin gelatin [Sigma, G7765] and 0.5% BSA, sterile filtered) for 30 min in a humidified chamber. Next, cells were incubated with the primary antibody for 30 min in a humidified chamber. Specific antibodies used and their respective dilutions can be found in Table EV1. Next, secondary antibody incubation in a dilution of 1:1000 for 30 min in a humidified chamber. DAPI nuclear staining (Sigma, D9564) was either included in the secondary antibody solution at a dilution of 1:5000 or after this incubation DAPI was prepared in PBS 1X at a dilution of 1:5000–1:10,000 and cells were incubated for 5 min. Coverslips were mounted using Mowiol (Calbiochem, 475904) on glass slides and air dried at room temperature protected from light until the next day. Next day the glass slides were either imaged in the microscope or stored at 4 °C for short- and long-term storage.

For immunohistochemistry coupled with smRNA FISH, same procedure as described for immunohistochemistry only, except the addition of DAPI which is included at the end of the smRNA FISH technique. After secondary antibody incubation, cells were quickly washed 3–4 times with PBS 1X and then fixed again with PFA 4% for 10 min. Then, cells were washed 3 times with PBS 1X. Next, for smRNA FISH we used Stellaris technology, whether commercial ready to use probes for *Ins1/2* transcripts (Biosearch Technologies, VSMF-3612-5), *LacZ* (Biosearch Technologies, VSMF-1002-5) or manually designed assays for *Ptprn*, *Pcsk1* and *Pcsk2* transcripts (Table EV4). First, 300 µL of buffer A were added to each well of a 24-well plate and incubated for 5 min at room temperature. Then, probes were prepared in hybridization buffer (HB [Biotechne,

SMF-HB1]) by adding 1 µl of probe to 1 mL of hybridization buffer (buffer had 900 µl of hybridization buffer and 100 µl of formamide [Roth, P040]). Cells were incubated in HB overnight at 37 °C in a parafilm-sealed humidified chamber. Next day, cells were washed once with buffer A for 30 min at 37 °C. Then, buffer A was removed, and next buffer A was added again but this time including DAPI in a dilution of 1:10,000 and again incubated for 30 min at 37 °C. Then, buffer A was removed and now buffer B was added for 5 min at room temperature. Finally, excess of buffer B was removed with a paper towel and then coverslips were mounted using Mowiol (Calbiochem, 475904) on glass slides. Glass slides were set to air dry at room temperature and protected from light overnight. Next day the glass slides were either imaged in the microscope or stored at 4 °C for short- and long-term storage.

## CRISPR-Cas9-mediated gene knockout

CRISPR-Cas9 system was utilized to generate G3BP1 and G3BP2 single knockout MIN6-K8 cell lines. Guidelines published to perform this technique were followed exactly as reported in (Ran et al, 2013). To generate the sgRNAs, the online tool from Integrated DNA Technologies (IDT). After transfections with the plasmids were performed, cells were taken to the Fluorescence-Activated Cell Sorting (FACS) facility to sort them based on GFP signal, sorting 1 cell per well in 96-well plates.

For rescue experiments, $G3BP1^{-/-}$ MIN6-K8 cell were transfected with pmCherry-G3BP1 construct (Trussina et al, 2024) (and after 3 days harvested for subsequent analyses by confocal microscopy, western blotting or FACS sorting using the red channel for detection of mCherry in a BD FACSAriaIII (BD Bioscience). mCherry$^+$ sorted cells were collected by centrifugation and insulin mRNA levels were quantified by qPCR as descriptive above.

## Insulin and proinsulin quantification by HTRF and ELISA

Cells were treated with resting and stimulation buffers as described before. After treatment, the buffer was separated in 3 aliquots and stored at −20 °C. Cells were harvested and pelleted as described before. Cell pellets were resuspended in lysis buffer and followed the same protocol for protein extraction mentioned before. Total protein extraction was also separated into 3 aliquots. To measure insulin and proinsulin levels, we used the Insulin Ultra-Sensitive Assay kit (Cisbio, 62IN2PEG) or the Rat/Mouse Proinsulin ELISA (Mercodia, 10-1232-01), respectively, as described by the manufacturers.

## ATP measurement and aldolase inhibition

ATP levels were measured with the CellTiter-Glo® 2.0 Cell Viability Assay (Promega, G9241), following the manufacturer's instructions. Relative Light Unit values were normalized to total protein levels as measured with the Pierce™ BCA Protein Assay Kit (ThermoFisher, 23225). The aldolase inhibitor Aldometanib (MedChemExpress, LXY-05-029), was used at different concentrations and buffers, as indicated in the Results section.

## Polysome profiling

Polysome profiling was performed as described in the article from (Aboulhouda et al, 2017). Briefly, cell lysates were prepared by

using specific reagents and buffer to ensure the preservation of ribosomes and mRNA integrity. Then, cell lysates were loaded onto sucrose gradients going from 10–50% of sucrose density and after the gradients were ultracentrifuged for 2 h and 30 min, to separate the polysomes based on their size, with heavier complexes migrating further. Finally, sucrose gradients were fractionated in an automated way by using the Piston Gradient Fractionator from Biocomp. A chart was obtained based on the absorbance at 260 nm of the sample revealing peaks for distinct components, such as 40S, 60S, 80S, and polysomes. Collected fractions were snap frozen with liquid nitrogen and then stored at $-80\,°C$.

## Mouse dispersed pancreatic islets and whole mount mouse pancreatic islets

Pancreatic islets were obtained from wild-type mice of the inbred C57JBL/6J strain. The protocol to prepare dispersed mouse pancreatic islets was adapted from (Phelps et al, 2017). Briefly, protocol begins with sterilizing and ECM-coating coverslips. Mouse islets are washed with PBS and digested using a mixture of Accutase and Cell Dissociation Solution at 37 °C. The digestion is halted with ice-cold FBS. The islets are then gently dispersed into a single-cell suspension and seeded onto the prepared coverslips. Following initial incubation at 37 °C, additional media is added. The cells are left undisturbed for several days to facilitate adhesion and spreading prior to subsequent experiments.

## Immunohistochemistry and smRNA FISH in mouse dispersed or whole pancreatic islets

Immunohistochemistry coupled with smRNA FISH was performed with the same protocol mentioned before for MIN6-K8 cells. A few details are different for dispersed and whole mount pancreatic islets, the glass coverslips used were of $24 \times 24$ mm (Roth, KCY2.1) and placed in 6-well plates (Corning, 353046). In addition, for the whole mount mouse pancreatic islets, permeabilization was performed with PBS-Triton 0.3% for 20 min.

## Immunohistochemistry and RNA scope in frozen human pancreatic tissue

Sections 10 μm human pancreatic cryosections were let to air dry for 5 min at room temperature, then a hydrophobic barrier was made by outlining the pancreatic tissue. Tissue was fixed for 15 min with PFA 4%, then 2 quick washes were performed with PBS 1X. Then, for INS RNA detection the protocol for RNAscope® Multiplex Fluorescent Reagent Kit v2 described by the manufacturer for frozen tissue sections was used. Subsequently, the tissue was again fixed with PFA 4% and washed twice with PBS 1X. Then, tissue was blocked with a drop of Dako antibody diluent (Dako, S0809) for 30 min. Then primary antibody for G3BP1 (Abcam, ab181150) was prepared in Dako antibody diluent in a 1:400 dilution. Tissue was washed 3 times with PBS 1X, then secondary antibody was prepared in Dako antibody diluent in a dilution of 1:1000 and including DAPI in a dilution of 1:10,000. Tissue was then washed 3 times with PBS 1X. Finally, 20 μl of mowiol were added on top of the tissue and then a $22 \times 22$ mm glass coverslip was placed on top of the mowiol drop trying to avoid bubble formation. Glass slides with the mounted coverslips were left to air dry at room temperature overnight. Next day, the glass slides were either imaged in the microscope or stored at 4 °C for short- and long-term storage.

## Image analysis

Confocal microscopy images were processed using Fiji software. Quantifications of count, area and size of G3BP1$^+$ condensates were performed by using the Analyze Particles function. Fluorescence intensity quantification was performed first by segmenting G3BP1$^+$ condensates with StarDist, and based on this segmentation, the *Ins1/2* mRNA fluorescence intensity was measured in its respective channel.

## Statistical analysis

Number of independent experiments are mentioned in each figure legend. All the statistical analyses were performed with GraphPad Prism Software. Tests such as Student's t-test and one-way ANOVA were indicated in each figure legend. Significant differences were depicted with stars for each graph and explained in the figure legends.

# Data availability

This study includes no data deposited in external repositories.

The source data of this paper are collected in the following database record: biostudies:S-SCDT-10_1038-S44318-025-00448-7.

# Peer review information

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

## Acknowledgements

We thank Dr. Janani Natarajan for helpful discussion, Dr. Titus M. Franzmann for the provision of the mCherry-hG3BP1 construct, Dr. Frank Möller for making the summary figure, Mrs. Katja Pfriem for administrative support, the Light Microscopy Facility a Core Facility of the CMCB Technology platform at TU Dresden and the CMCB Flow Cytometry Core Facility. Work in the Solimena was supported by the German Center for Diabetes Research (DZD e.V.), which is financed by the German Ministry for Education and Research; the DFG Grant SO 818/10-1; the European Union Funded Network "INTERCEPT-T2D" 101095433; the Innovative Medicines Initiative 2 Joint Undertaking under grant agreement 115881 (RHAPSODY), which includes financial contributions from the European Union's Framework Program Horizon 2020, the European Federation of Pharmacological Industries and Associations (EFPIA), and the Swiss State Secretariat for Education, Research and Innovation under contract 16.0097. Views and opinions expressed are, however, those of the authors only and don't necessarily reflect those of the funding agencies. Neither the European Union nor any of the granting authorities can be held responsible for them.

## Author contributions

**Esteban Quezada**: Conceptualization; Data curation; Software; Formal analysis; Supervision; Validation; Investigation; Visualization; Methodology; Writing—original draft; Project administration; Writing—review and editing. **Klaus-Peter Knoch**: Conceptualization; Validation; Investigation; Methodology; Writing—review and editing. **Jovana Vasiljevic**: Conceptualization; Formal analysis; Funding acquisition; Methodology. **Annika Seiler**: Data curation; Formal analysis; Investigation; Methodology. **Akshaye Pal**: Investigation. **Abishek Gunasekaran**: Conceptualization; Investigation; Methodology. **Carla Münster**: Investigation. **Daniela Friedland**: Investigation. **Eyke Schöniger**: Investigation. **Anke Sönmez**: Investigation. **Pascal Roch**: Formal analysis; Investigation; Methodology. **Carolin Wegbrod**: Investigation. **Katharina Ganß**: Investigation. **Nicole Kipke**: Investigation. **Simon Alberti**: Supervision. **Rita Nano**: Resources. **Lorenzo Piemonti**: Resources. **Daniela Aust**: Resources. **Jürgen Weitz**: Resources. **Marius Distler**: Resources. **Michele Solimena**: Conceptualization; Formal analysis; Supervision; Funding acquisition; Project administration; Writing—review and editing.

Source data underlying figure panels in this paper may have individual authorship assigned. Where available, figure panel/source data authorship is listed in the following database record: biostudies:S-SCDT-10_1038-S44318-025-00448-7.

## Funding

## Disclosure and competing interests statement

The authors declare no competing interests.

# Expanded View Figures

## Resting - 2.8 mM Glucose

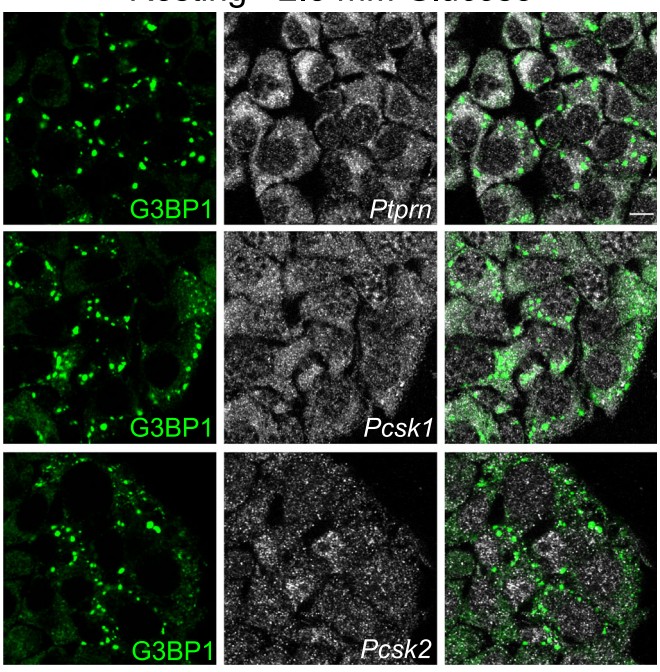

**Figure EV1.  mRNA detection of other insulin secretory granule cargoes.**

(A) Immunostainings for G3BP1 (green) and smRNA FISH for *Ptprn*, *Pcsk1* and *Pcsk2* (gray) in MIN6-K8 cells in resting glucose conditions.

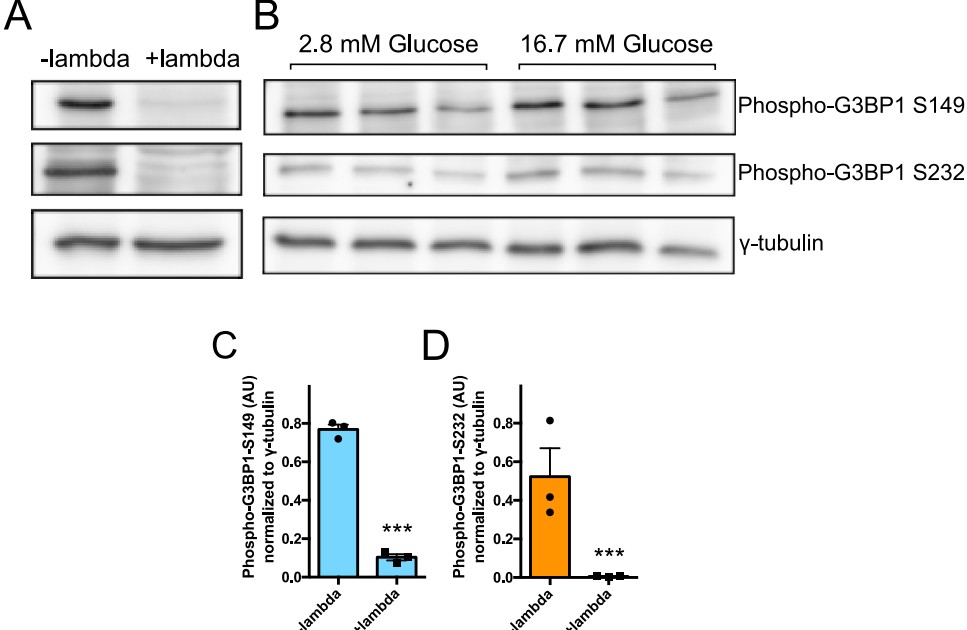

**Figure EV2. Western blots for phospho-G3BP1 S149 and phospho-G3BP1 S232.**

(A) Western blot for Phospho-G3BP1 S149, Phospho-G3BP1 S232 and gamma-tubulin on extracts of MIN6-K8 cells in growth media with 25 mM glucose and treated with or without lambda phosphatase. (B) Western blot for Phospho-G3BP1 S149, Phospho-G3BP1 S232 and gamma-tubulin on extracts of glucose resting or stimulated MIN6-K8 cells. (C, D) Quantification of the western blots shown in panel (A). (C, D): Every comparison $p < 0.0001$. Data information: Presented values denote the mean ± SD derived from three independent experiments, analyzed via paired *t*-test with Mann–Whitney correction. Values with ***$p < 0.001$ were considered statistically significant relative to the -lambda condition. The results for the western blot are from 1 technical replicate of each condition per independent experiment.

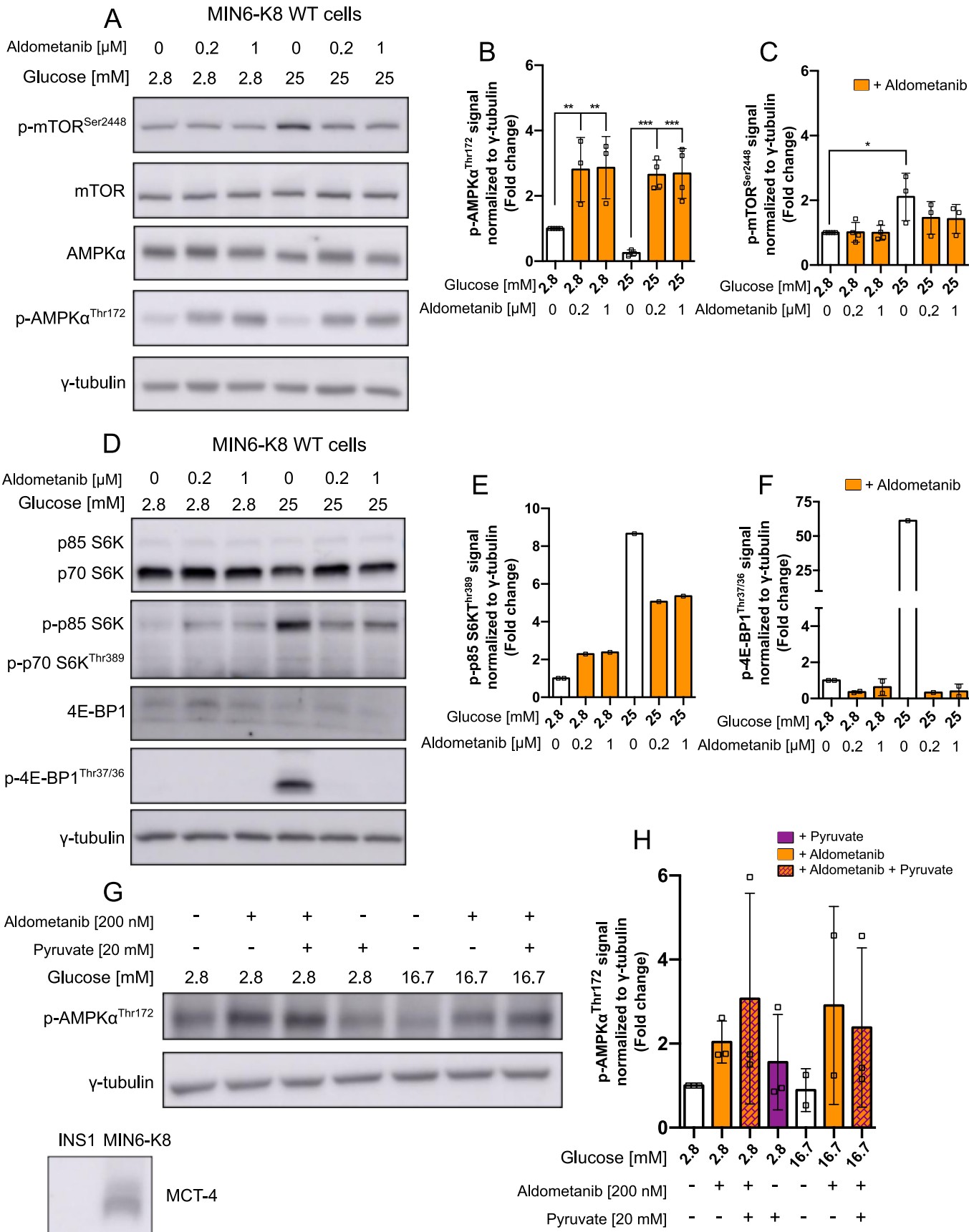

◀ **Figure EV3. Aldolase control of AMPKα and mTOR signaling in MIN6-K8 cells.**

(A) Western blot (A) and quantifications (B) p-AMPKα$^{Thr172}$, (C) p-mTOR$^{Ser2448}$ and gamma-tubulin across varying conditions: resting and stimulating glucose concentrations, with and without aldometanib treatment as shown in the figure. (D) Western blot (D) and quantifications of (E) p-p85 S6K$^{Thr389}$, (F) p-4E-BP1$^{Thr37/36}$ and gamma-tubulin across varying conditions: resting and stimulating glucose concentrations, with and without aldometanib treatment as shown in the figure. (G) Western blot (G) and quantifications (H) p-AMPKα$^{Thr172}$ and gamma-tubulin across varying conditions: resting and stimulating glucose concentrations, with and without aldometanib treatment and/or pyruvate as shown in the figure. Data information: Presented values denote the mean ± SD derived from three independent experiments (B, C, H) or two independent experiments (E, F), analyzed via paired $t$-test with Mann–Whitney correction. Values with **$p < 0.01$ and ***$p < 0.001$ were considered statistically significant relative to the conditions as shown in each graph. The results of the western blots are from 2 technical replicates of each condition per independent experiment. (B): 2.8 vs 2.8 + aldometanib 0.2 ($p = 0.0043$); 2.8 + aldometanib 1 ($p = 0.0035$), 25 vs 25 + aldometanib 0.2 ($p = 0.0003$); 25 + aldometanib 1 ($p = 0.0007$). (C): 2.8 vs 25 ($p = 0.0319$).

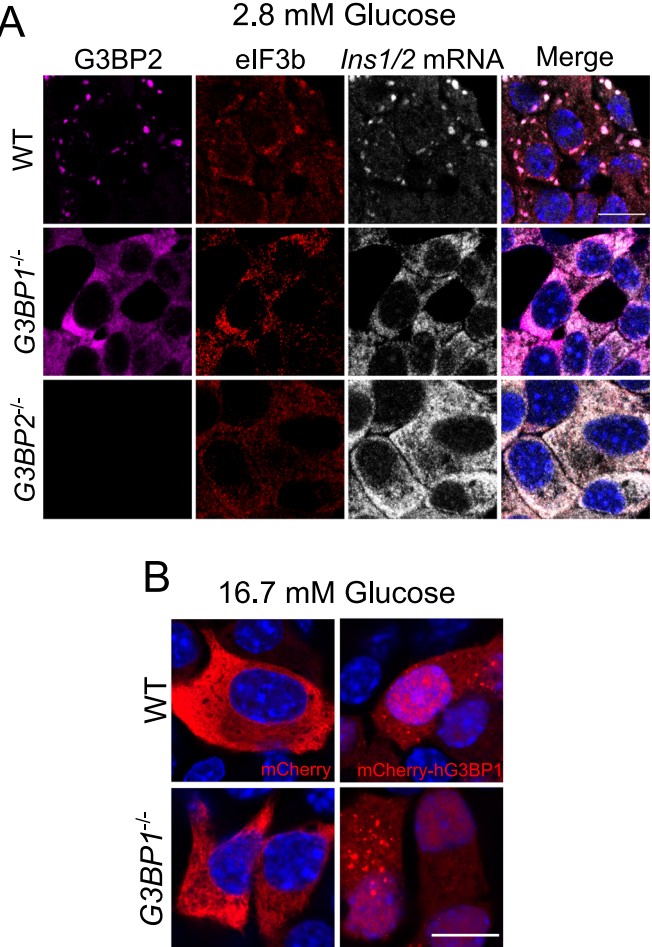

**Figure EV4.  Confocal microscopy of *G3BP1⁻ᐟ⁻*, *G3BP2⁻ᐟ⁻* and wild-type MIN6-K8 cells.**

(**A**) Immunostaining for G3BP2 (magenta), eIF3b (red) and smRNA FISH for *Ins1/2* in WT, *G3BP1⁻ᐟ⁻* and *G3BP2⁻ᐟ⁻* MIN6-K8 cells at resting glucose concentrations. Nuclei are stained with DAPI (blue). (**B**) Immunostaining of *G3BP1⁻ᐟ⁻* and wild-type MIN6-K8 cells transiently transfected with *mCherry* or *mCherry-hG3BP1* and exposed to high glucose levels. Signals for mCherry or mCherry-hG3BP1 are in red, as labeled in the image. Nuclei are stained with DAPI (blue). Data information: The presented results in (**A**) were derived from three independent experiments and at least 5 images per condition per experiment, and for (**B**) one independent experiment and 4 images of each condition. Scale bars = 10 μm.

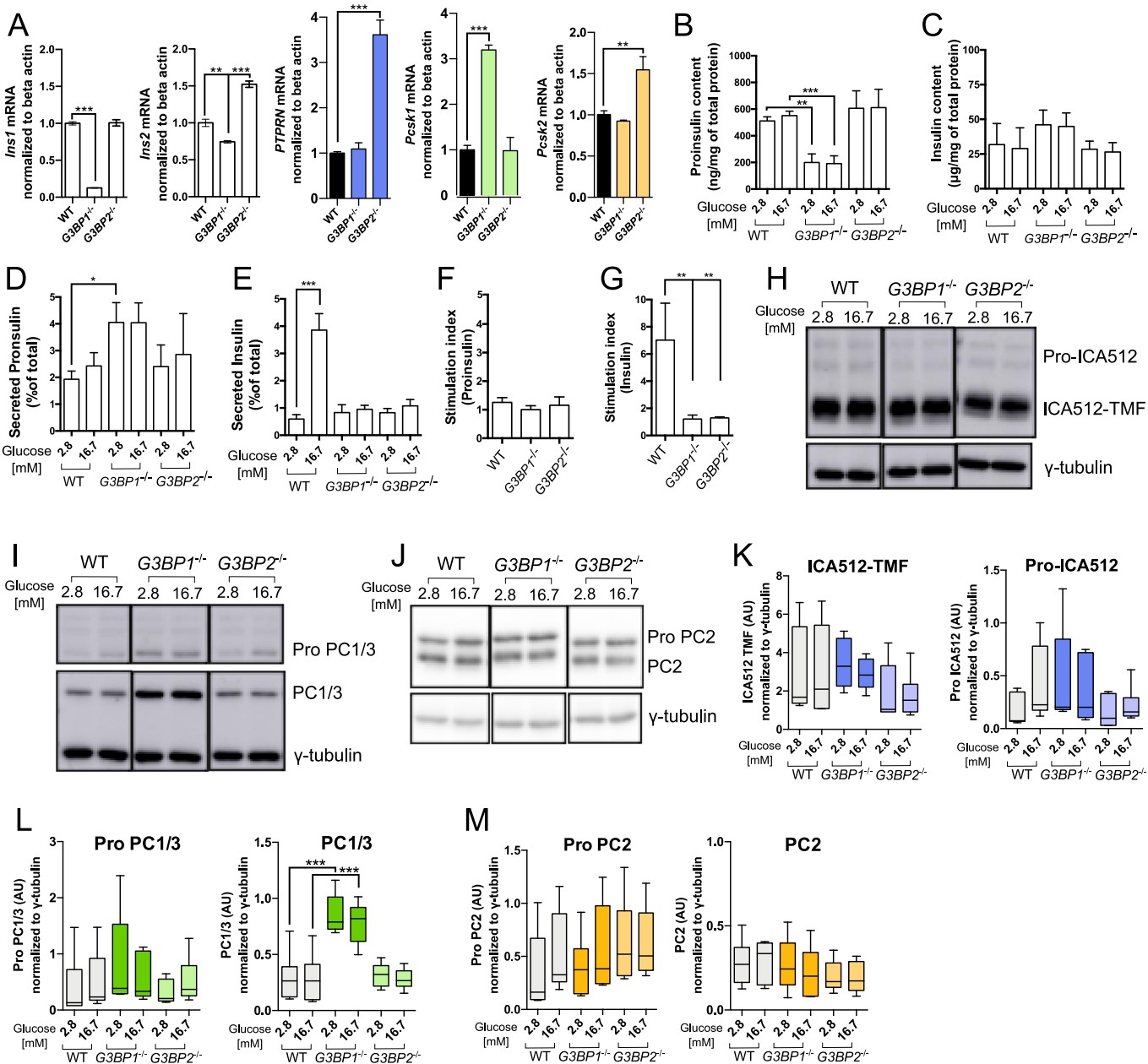

**Figure EV5. mRNA and protein levels of insulin secretory granule cargoes in MIN6-K8 wild type, *G3BP1*$^{-/-}$ and *G3BP2*$^{-/-}$ cells.**

(A) *Ins1*, *Ins2*, *Ptprn*, *Pcsk1*, and *Pcsk2* mRNA levels in WT, *G3BP1*$^{-/-}$ and *G3BP2*$^{-/-}$ MIN6-K8 cells, as assessed by qRT-PCR. (B–E) Quantification of proinsulin and insulin levels and their secretion to the culture media as measured by ELISA and HTRF, respectively, in WT, *G3BP1*$^{-/-}$ and *G3BP2*$^{-/-}$ MIN6-K8 cells under resting and stimulating glucose concentrations. (F, G) Stimulation index for proinsulin or insulin, respectively, in WT, *G3BP1*$^{-/-}$ and *G3BP2*$^{-/-}$ MIN6-K8 cells. (H–M) Western blots and their respective quantifications for Pro-ICA512, ICA512 Transmembrane Fragment (ICA512-TMF), Pro-PC1/3, PC1/3, Pro-PC2 and PC2 species in glucose resting and stimulated WT, *G3BP1*$^{-/-}$ and *G3BP2*$^{-/-}$ MIN6-K8 cells. Data information: Presented values denote the mean ± SD derived from three independent experiments for qRT-PCR and five independent experiments for western blots, analyzed via one-way ANOVA. Values **$p < 0.01$, and ***$p < 0.001$ were considered statistically significant relative to the WT condition. The results for the qRT-PCR are from 3 technical replicates and for western blot from 1 technical replicate of each condition per independent experiment. (A): *Ins1* graph WT vs *G3BP1*$^{-/-}$ ($p < 0.0001$), *Ins2* graph WT vs *G3BP1*$^{-/-}$ ($p = 0.0038$); *G3BP2*$^{-/-}$ ($p < 0.0001$), *PTPRN* graph WT vs *G3BP2*$^{-/-}$ ($p < 0.0001$), *Pcsk1* graph WT vs *G3BP1*$^{-/-}$ ($p < 0.0001$), *Pcsk2* graph WT vs *G3BP2*$^{-/-}$ ($p = 0.008$). (B): 2.8 WT vs 2.8 *G3BP1*$^{-/-}$ ($p = 0.0012$), 16.7 WT vs 16.7 *G3BP1*$^{-/-}$ (0.0009). (D): 2.8 WT vs 2.8 *G3BP1*$^{-/-}$ ($p = 0.029$). (E): 2.8 WT vs 16.7 WT ($p < 0.0001$). (G): WT vs *G3BP1*$^{-/-}$ ($p = 0.0041$); vs *G3BP2*$^{-/-}$ ($p = 0.0037$). (L): PC1/3 graph 2.8 WT vs 2.8 *G3BP1*$^{-/-}$ ($p < 0.0001$), 16.7 WT vs 16.7 *G3BP1*$^{-/-}$ ($p < 0.0001$).

# A

### G3BP1 cDNA

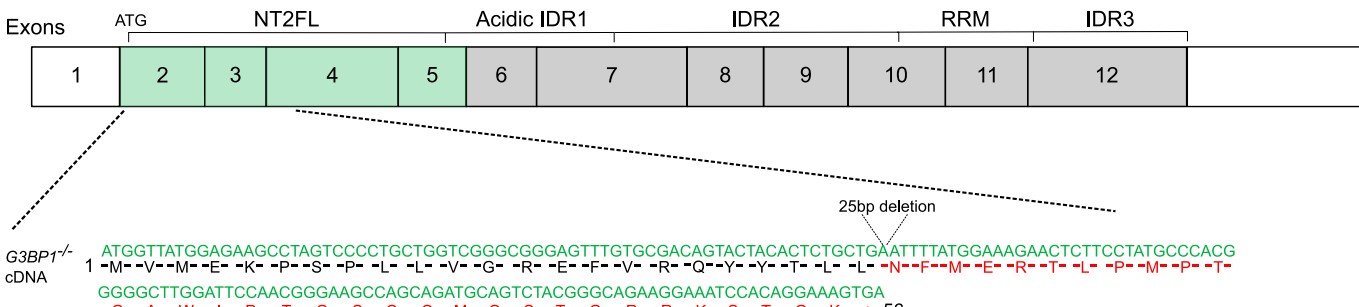

# B

### G3BP2 cDNA

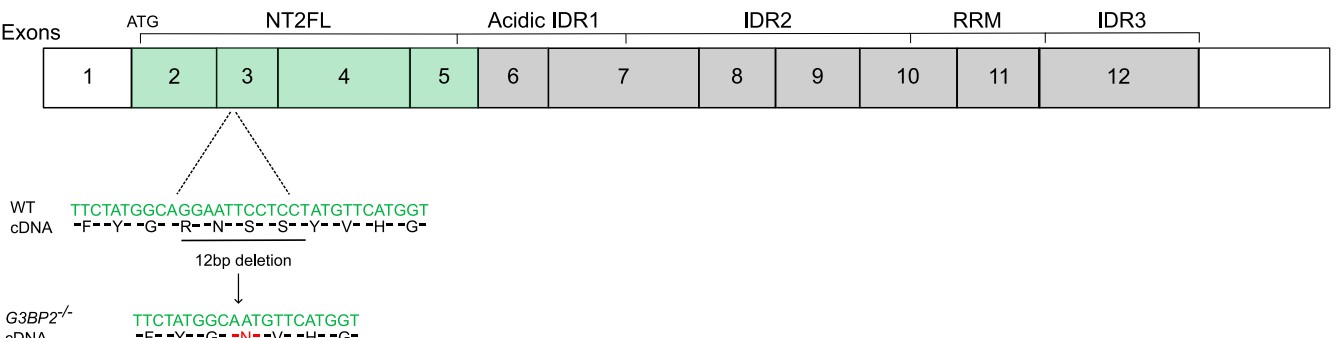

**Figure EV6.   Characterization of *G3BP1* and *G3BP2* deletions in *G3BP1*⁻/⁻ and *G3BP2*⁻/⁻ MIN6-K8 clones.**

(A, B) Schematic illustrations of the G3BP1 (A) and G3BP2 (B) domain and exon structures. The exons coding for the NTF2L domain, which is responsible for G3BP1 and G3BP2 dimerization are colored in green. The location of the nucleotide deletions identified in *G3BP1*⁻/⁻ and *G3BP2*⁻/⁻ MIN6-K8 clones and the alterations in the corresponding amino sequences are shown. The deletion in *G3BP1* introduces a premature stop codon. The deletion in *G3BP2* removes four amino acids and converts Y40 into N.

