## [Peer Review File · The EMBO Journal]

Aldolase-regulated G3BP1/2+ condensates control insulin mRNA storage in beta cells

Esteban Quezada, Klaus-Peter Knoch, Jovana Vasiljević, Annika Seiler, Akshaye Pal, Abishek Gunasekaran, Carla Münster, Daniela Friedland, Eyke Schöniger, Anke Sönmez, Pascal Roch, Carolin Wegbrod, Katharina Ganß, Nicole Kipke, Simon Alberti, Rita Nano, Lorenzo Piemonti, Daniela Aust, Jürgen Weitz, Marius Distler, and Michele Solimena

Corresponding author(s): Michele Solimena (michele.solimena@tu-dresden.de)

Review Timeline:

Submission Date:	22nd May 24
Editorial Decision:	4th Sep 24
Revision Received:	14th Jan 25
Editorial Decision:	12th Feb 25
Revision Received:	13th Mar 25
Accepted:	2nd Apr 25

Editor: Cornelius Schneider

Transaction Report:

Dear Dr. Solimena,

Thank you for submitting your manuscript for consideration by the EMBO Journal, for sharing a preliminary revision plan with me and also for the productive meeting.

As discussed in the meeting, we find the proposed revision experiments reasonable and adequate to address the concerns raised by both referee #1 and #2 and we will not require that you perform additional in vivo mouse experiments asked by referee #3 in point 3 and 4.

Based on these considerations and your willingness to engage in a major revision as indicated during the pre-decision consultation, I would like to invite you to submit a revised version of the manuscript, addressing the comments of all three reviewers. I should add that it is EMBO Journal policy to allow only a single round of revision, and acceptance of your manuscript will therefore depend on the completeness of your responses in this revised version. If you have any additional questions or want to discuss the revisions further, I am happy to do so by email or video conferencing.

We generally allow three months as standard revision time, which can be extended to 6 months in case of major revisions, such as the experiments required here. As a matter of policy, competing manuscripts published during this period will not negatively impact on our assessment of the conceptual advance presented by your study. However, we request that you contact the editor as soon as possible upon publication of any related work, to discuss how to proceed. Should you foresee a problem in meeting the deadline, please let us know in advance and we may be able to grant an extension.

Thank you for the opportunity to consider your work for publication. I look forward to your revision.

Yours sincerely,

Cornelius Schneider, PhD
Editor
The EMBO Journal
c.schneider@embojournal.org

- a Reagents and Tools Table as part of the Methods section, which can be downloaded from our author guidelines

(<https://www.embopress.org/page/journal/14602075/authorguide#structuredmethods>)

We realize that it is difficult to revise to a specific deadline. In the interest of protecting the conceptual advance provided by the work, we recommend a revision within 3 months (3rd Dec 2024). Please discuss the revision progress ahead of this time with the editor if you require more time to complete the revisions. Use the link below to submit your revision:

Referee #1:

The manuscript by Quezada and collaborators examines the potential role of G3BP1/2 proteins in regulating the insulin mRNAs in beta cells. The main observation is that in low glucose conditions the G3BP1/2 proteins and the Insulin mRNAs form RNP granules, which then disperse with glucose addition. This is an interesting and clear observation. The impact of these RNP granules remains unclear (issues detailed below). As such, I think the work could be appropriate for EMBO provided the issues below were well addressed.

This review is from Roy Parker and I would be willing to clarify these comments for the authors if needed.

Specific Comments:

1) The simplest explanation for the glucose sensitive formation of RNP granules is that low glucose leads to strongly reduced translation, which is then increased upon glucose restoration. While this is consistent with the polysomes, I would suggest a simple cell labeling with puromycin to examine if there is a global change in translation rates (PMID: 19305406) with high and low glucose.

(It might also be useful to examine such a change in the *g3bp1Δ* and *g3bp2Δ* cells, if the translation defect in *g3bp1Δ* cells is truly dependent on G3BP1 (see below)).

2) A key part of the manuscript are the changes in insulin mRNAs and protein production in the *g3bp1Δ* and *g3bp2Δ* cell lines. Since we often observe off-target effects of CRISPR genetic changes, it would be important to demonstrate the phenotypes are rescued by introduction of a G3BP1 transgene (ideally stably). Reintroducing G3BP1 into the G3BP null cell line should rescue the pro-insulin levels and confirm the results are not due to off-target effects.

Similarly, as it stands now, the polysome analyses argue that G3BP1 is required for resuming translation in response to glucose. It would be important to determine if that defect in translation is complemented by a G3BP1 transgene.

3) It was unclear to me how G3BP1 can reduce pro-insulin production but not insulin (in figure 3 F, G and I). One explanation could be the timing and source of the specific samples being analyzed (media or cells?). Can the authors a) provide more information on the specifics of this experiment and/or b) provide a possible molecular explanation?

4) The manuscript would be more impactful if the authors determined whether G3BP1 and/or G3BP2 altered the secretion of insulin protein in response to a glucose feeding. This seems the critical biological issue and it was not clear to me this experiment was performed in the work. [Or am I wrong, and the production of insulin is independent of glucose addition (as suggested by the results in 3F)]. This experiment would distinguish whether the observed RNP granules are simply assemblies that are forming during translation repression due to low glucose, or are critical for the biological response to glucose.

Minor comments -

5) It would be interesting to know if the G3BP + Ins1/2 granules are enriched in the same RNAs as canonical stress granules (AHNAK or DYNC1H1 for example). Understanding how similar or different these granules are to stress granules could help determine the nature of these assemblies.

6) On page 7 the authors write "G3BP2 may be implicated in Ins2 mRNA degradation." In principle, the change in RNA level could be due to either differences in transcription and/or degradation. They should either be specific about both possibilities and/or examine the stability following actinomycin D treatment.

7) Please number the figures. The lack of numbering of the figures makes the manuscript difficult to follow. Also, the subfigure numbering in the text for figure 1 is incorrect (pg.5)

8) The y axis titles for some bar graphs (see figure 2 B and C) are confusing. The title reads "average number of..." or "average size of..." which indicates that the points are averages of experiments. Due to the number of points, I presume they are, instead, the absolute numbers for each cell that makes up the experiments. I recommend removing the word "average" from the axis title.

Referee #2:

In this interesting manuscript, the authors use a glucose-responsive insulinoma mouse cell line to investigate the role of G3BP1 and G3BP2 in regulating *Ins1/2* mRNA upon glucose stimulation. The study follows the previous discovery that G3BP1 is one of the most downregulated mRNAs in islets from type 2 diabetes patients, raising the question of the functional significance. They found that G3BP1 and G3BP2 form condensates at low glucose levels (corresponding to fasting), which dissipate when the glucose concentration is increased or after treatment with drugs that induce insulin synthesis. These condensates contain *Ins1/2* mRNA, but not mRNA of other genes involved in insulin secretion. They convincingly show that the condensates depend on either G3BP1 or 2 to be formed, but that only deletion of G3BP1 induces a decrease in *Ins1/2* mRNA and corresponding protein levels. In the last part of the paper, they show that similar G3BP1 condensates containing *Ins1/2* mRNA can be found in beta cells from mouse or human non diabetic individuals.

Overall, this is an interesting study that suggests that G3BP1 and 2 may regulate the translation or the stability of *Ins1/2* mRNA by forming membraneless condensates, bearing some but not complete similarity to stress granules (SGs). However, the manuscript lacks the final steps to demonstrate the role of the condensates in insulin translation and/or mRNA protection from degradation.

Major comments:

1. The authors show data suggesting that the G3BP1/2 + condensates at low glucose concentrations are partially distinct from canonical SGs. Are these condensates resolved by treatment with cycloheximide?
2. The authors' hypothesis, well justified by their data and the available literature on SGs, is that the condensates are sites of translation inhibition and/or mRNA protection. However, there is no formal experimental evidence to support this hypothesis. The authors could easily test whether *Ins1/2* mRNA is indeed highly translated in high glucose, but not in the granules, and how the lack of G3BP1 affects translation of this mRNA, using for example puromycin labelling coupled with proximity ligation (Puro-PLA technique). The polysome experiment does not answer this question.
3. Related to the point above, it would be important to test whether the stability of *Ins1/2* mRNA is indeed reduced in the G3BP1 KO cells.
4. As a note of caution, the GAPDH sm-RNA FISH signal appears to be very diffuse, which casts doubt on its specificity. One would expect to see single spots in such an experiment. A negative control for the FISH experiments would be appropriate. For example, using a probe against a gene that is not expressed in human cells to validate how much background is present in these experiments. I could not find in the methods how the GAPDH probe was prepared, but it seems that a commercial probe was used for the *Ins1/2* mRNAs, as opposed to home-made probes for the other mRNAs. This may of course affect the quality of the signal differently.

Minor comments:

1. In Figure 5A, B I have the impression that the condensates appear particularly in cells with low levels of insulin. Is this because of the relation between the condensates and the expression of insulin? The data are not totally convincing, as there are many cells positive for insulin in which G3BP1 does not form granules. The authors should comment on these findings.
2. The blot showing phospho-AMPK is not really striking. The authors discuss extensively about the different behavior of the granules that they see in regard to phosphorylation of AMPK, but then there is no experiment to further investigate the signalling cascade involved. The discussion on the difference between these granules and SGs should be downplayed, unless there are more experiments done in this regard.
3. Figure 3F: it seems that the figure contains the merge of two separate blots. If this is the case it should be indicated. Comparison can be done only if some lanes on the same blot were spliced out.

Referee #3:

In this manuscript, Esteban Quezada et al found that *Ins1/2* mRNA is enriched in G3BP1+/G3BP2+/eIF3b+ condensates in resting MIN6-K8 cells, while MIN6-K8 cells stimulated with 16.7 mM glucose, *ins*/G3BP1/G3BP2 were redistributed throughout the cytosol and their condensates were no longer detectable. These are interesting phenomenon. Understanding, the mechanism of *ins*/G3BP1/G3BP2 cytosolic condensates formation is still confused. Why only *ins1/2* is detectable in G3BP1+condensates of resting MIN6-K8 cells, while *Ptprn*/ICA512, *Pcsk1* and *Pcsk2* were not detected. This reduces my overall enthusiasm. Overall, the data included in the paper are interesting. However, there are several points in the manuscript

that in my opinion need to be improved or clarified.

Major comments

(1) Based on Wigger et al. (2021) study, G3BP1 was significantly downregulated in the islets of living donors with T2D compared to normoglycemic individuals, while author found that the mRNA and protein almost have no change in the MIN6-K8 cells induced by glucose, why? Palmitate or other factors would affect G3BP1 expression? The expression level of G3BP1 in the islets of T2DM model mice need investigate?

(2) Why only ins1/2 is detectable in G3BP1+ condensates of resting MIN6-K8 cells, while Ptprn/ICA512, Pcsk1 and Pcsk2 were not detected. The precise mechanism of G3BP1+ / and G3BP2+ /Ins1/2 mRNA+ condensates formation needs to further research.

(3) More deep analysis of G3BP1+ / and G3BP2+ /Ins1/2 mRNA+ condensates in the islets of T2DM model mice (such as HFD, db/db and ob/ob) should be include.

(4) since different glucose level effect G3BP1+ / and G3BP2+ /Ins1/2 mRNA+ condensates, what is trend of G3BP1+ / and G3BP2+ /Ins1/2 mRNA+ condensates in the development of T2DM, please investigate it in the islets of mice.

(5) G3BP1 and G3BP2 are critical constituents of stress granules, and the mechanism is various (doi:

10.1016/j.cell.2020.03.046. doi: 10.1126/science.abf6548.). why author think G3BP1 and G3BP2 condensates regulated via AMPK α pathway in the islets, it seems too casual. Overall, the mechanism of ins1/2/G3BP1/G3BP2 cytosolic condensates formation is still confused. I would like to see a mRNA or protein analyses to show which genes are likely to be directly regulated by G3BP1 and G3BP2 in the MIN6-K8 cells.

(6) Figure 2D showed that AMPK α was decreased, The relationship between G3BP1, and AMPK α is not clear and must be clarified.

(7) Resting condition in Figure2-4 is 2.8 mM glucose. Resting condition in Figure 5 choose 2.8 mM and 3.3 mM glucose, please clarify the reason.

(8) It is better to investigate this phenomenon that ins1/2/G3BP1/G3BP2 cytosolic condensates formation in the human β cell line (such as Endoc- β H1)

Minor comments

(1) please add the animal ethics committee in the methods of mice.

(2) As sample size for each experiment should be included. Also, details related to any replicates/repetitions can be included.

(3) Quantifying the western blot images in Figure 3 and 2 would enhance the readability.

The manuscript by Quezada and collaborators examines the potential role of G3BP1/2 proteins in regulating the insulin mRNAs in beta cells. The main observation is that in low glucose conditions the G3BP1/2 proteins and the Insulin mRNAs form RNP granules, which then disperse with glucose addition. This is an interesting and clear observation. The impact of these RNP granules remains unclear (issues detailed below). As such, I think the work could be appropriate for EMBO provided the issues below were well addressed. This review is from Roy Parker and I would be willing to clarify these comments for the authors if needed.

Specific Comments:

1) The simplest explanation for the glucose sensitive formation of RNP granules is that low glucose leads to strongly reduced translation, which is then increased upon glucose restoration. While this is consistent with the polysomes, I would suggest a simple cell labeling with puromycin to examine if there is a global change in translation rates (PMID: 19305406) with high and low glucose.

As suggested by Dr. Parker, changes in global translation in response to glucose were examined by puromycin labeling. These results, which are shown in the new Fig. 4L-M, corroborate that glucose stimulation rapidly upregulates global translation of beta cells.

(It might also be useful to examine such a change in the *g3bp1Δ* and *g3bp2Δ* cells, if the translation defect in *g3bp1Δ* cells is truly dependent on G3BP1 (see below).

2) A key part of the manuscript are the changes in insulin mRNAs and protein production in the *g3bp1Δ* and *g3bp2Δ* cell lines. Since we often observe off-target effects of CRISPR genetic changes, it would be important to demonstrate the phenotypes are rescued by introduction of a G3BP1 transgene (ideally stably). Reintroducing G3BP1 into the G3BP null cell line should rescue the pro-insulin levels and confirm the results are not due to off-target effects. Similarly, as it stands now, the polysome analysis argues that G3BP1 is required for resuming translation in response to glucose. It would be important to determine if that defect in translation is complemented by a G3BP1 transgene.

We appreciate the suggestion to confirm the specificity of our results and to address potential off-target effects of CRISPR-mediated deletions. To this end, we performed experiments in which we transiently expressed the mCherry-hG3BP1 construct in *G3BP1*^{-/-} MIN6-K8 cells (new Fig. 4J). Expression of mCherry-hG3BP1 effectively rescued the phenotype observed in these cells. Specifically, the presence of G3BP1⁺ condensates in resting glucose was restored (new Fig. 4D), proinsulin content returned to levels in wild type MIN6-K8 cells (new Fig. 4J), while *Ins1* and *Ins2* mRNA expression partially recovered, their levels being approximately 50% and 200% compared to wild-type cells, respectively (new Fig. 4H). These results indicate that the phenotypes observed in *G3BP1*^{-/-} cells are G3BP1-dependent and not the result of off-target effects.

3) It was unclear to me how G3BP1 can reduce pro-insulin production but not insulin (in figure 3 F, G and I). One explanation could be the timing and source of the specific samples being

analyzed (media or cells?). Can the authors a) provide more information on the specifics of this experiment and/or b) provide a possible molecular explanation?

Dr. Parker is correct. The previous data about insulin secretion from wt MIN6-K8 cells were not reliable, conceivably due to the use of a late passage of the cells. We performed again all the measurements but with wt cells from earlier passages, which properly secrete insulin when stimulated with high glucose (new Fig. S2E and S2G). We found that there is indeed an impairment in glucose stimulated insulin secretion in both *G3BP1*^{-/-} and *G3BP2*^{-/-} MIN6-K8 cells (new Fig. S2E and S2G). Reduction in proinsulin content in *G3BP1*^{-/-} cells remained consistent with previous results. Higher proinsulin secretion from resting *G3BP1*^{-/-} MIN6-K8 could partially account for the reduction in proinsulin content (new Fig. S2B and S2D) .

4) The manuscript would be more impactful if the authors determined whether G3BP1 and/or G3BP2 altered the secretion of insulin protein in response to a glucose feeding. This seems the critical biological issue and it was not clear to me this experiment was performed in the work. [Or am I wrong, and the production of insulin is independent of glucose addition (as suggested by the results in 3F)]. This experiment would distinguish whether the observed RNP granules are simply assemblies that are forming during translation repression due to low glucose, or are critical for the biological response to glucose.

As indicated above, we now found that both *G3BP1*^{-/-} and *G3BP2*^{-/-} MIN6-K8 cells display impaired glucose stimulated insulin secretion.

Minor comments –

5) It would be interesting to know if the G3BP + Ins1/2 granules are enriched in the same RNAs as canonical stress granules (AHNAK or DYNC1H1 for example). Understanding how similar or different these granules are to stress granules could help determine the nature of these assemblies.

We evaluated a probe set targeting *DYNC1H1* mRNA but unfortunately we could not detect any signal. This negative result may reflect a technical problem, despite having followed the guidelines for custom probe design from Stellaris and also ordered a new *Ins1/2* probe set, which worked properly. As negative control, we included a probe set targeting LacZ (new Fig. 1E-F).

6) On page 7 the authors write "G3BP2 may be implicated in Ins2 mRNA degradation." In principle, the change in RNA level could be due to either differences in transcription and/or degradation. They should either be specific about both possibilities and/or examine the stability following actinomycin D treatment.

We thank Dr. Parker for this suggestion. We analyzed *G3BP2*^{-/-} cells and wt MIN6-K8 cells upon treatment with actinomycin D for up to 24 hours and found that in *G3BP2*^{-/-} MIN6-K8 cells, unlike in wt cells, the levels of the *Ins1* mRNA variant were not reduced, suggesting that G3BP2 may indeed be implicated in *Ins1* mRNA degradation. The results are shown in the new Fig. 4F.

7) Please number the figures. The lack of numbering of the figures makes the manuscript difficult to follow. Also, the subfigure numbering in the text for figure 1 is incorrect (pg.5)

Many thanks. These issues have been corrected.

8) The y axis titles for some bar graphs (see figure 2 B and C) are confusing. The title reads "average number of..." or "average size of..." which indicates that the points are averages of experiments. Due to the number of points, I presume they are, instead, the absolute numbers for each cell that makes up the experiments. I recommend removing the word "average" from the axis title.

Dr. Parker is correct. The labels of the y-axis in Fig. 2C-D have been corrected.

Reviewer #2

1) The authors show data suggesting that the G3BP1/2 + condensates at low glucose concentrations are partially distinct from canonical SGs. Are these condensates resolved by treatment with cycloheximide?

We thank the reviewer for suggesting this analysis. As shown in the new Fig. 2A-D, treatment of resting MIN6-K8 cells with cycloheximide resolves G3BP1/2 + condensates.

2) The authors' hypothesis, well justified by their data and the available literature on SGs, is that the condensates are sites of translation inhibition and/or mRNA protection. However, there is no formal experimental evidence to support this hypothesis. The authors could easily test whether *Ins1/2* mRNA is indeed highly translated in high glucose, but not in the granules, and how the lack of G3BP1 affects translation of this mRNA, using for example puromycin labelling coupled with proximity ligation (Puro-PLA technique). The polysome experiment does not answer this question.

As also suggested by reviewer #1, we have labeled MIN6-K8 cells with puromycin and corroborated that glucose stimulation prompts their global translation (Fig. 4L-M). These data further show (and thus of G3BP1/2⁺ condensates), global translation is upregulated in resting G3BP1^{-/-} MIN6-K8 cells compared to wt cells. Due to time constraints we could not couple the puromycin labelling with proximity ligation. However, in a previous study we found that pull down assays with the 5'-UTR of *insulin2* mRNA, which is the most abundant insulin species in mouse beta cells, resulted in the recovery of G3BP1 from cytosolic extracts of both resting and glucose stimulated MIN6 cells (Fig. 1E-F; Vasiljević et al, BioRxiv, 2021; <https://doi.org/10.1101/2021.05.07.443159>).

3) Related to the point above, it would be important to test whether the stability of *Ins1/2* mRNA is indeed reduced in the G3BP1 KO cells.

See comments our reply to point 6) in the minor comments of Reviewer #1

4. As a note of caution, the GAPDH sm-RNA FISH signal appears to be very diffuse, which casts doubt on its specificity. One would expect to see single spots in such an experiment. A

negative control for the FISH experiments would be appropriate. For example, using a probe against a gene that is not expressed in human cells to validate how much background is present in these experiments. I could not find in the methods how the GAPDH probe was prepared, but it seems that a commercial probe was used for the *Ins1/2* mRNAs, as opposed to home-made probes for the other mRNAs. This may of course affect the quality of the signal differently.

Following the suggestion of the reviewer, we repeated the smRNA FISH with a probe from Stellaris targeting LacZ of *Escherichia coli* as negative control. These results are shown in the new Fig. 1E-F.

Minor comments:

1. In Figure 5A, B I have the impression that the condensates appear particularly in cells with low levels of insulin. Is this because of the relation between the condensates and the expression of insulin? The data are not totally convincing, as there are many cells positive for insulin in which G3BP1 does not form granules. The authors should comment on these findings.

While we did not quantify the number/area of G3BP1⁺/2⁺ condensates in relationship to the intensity of the insulin signal we also share the impression that the two are inversely correlated.

2. The blot showing phospho-AMPK is not really striking. The authors discuss extensively about the different behavior of the granules that they see in regard to phosphorylation of AMPK, but then there is no experiment to further investigate the signalling cascade involved. The discussion on the difference between these granules and SGs should be downplayed, unless there are more experiments done in this regard.

We have now added new results about the AMPK α -aldolase axis in relationship to condensates (PMID: 29153408, PMID: 25840010) (revised Fig. 2E-G). We found that alogliptin-mediated inhibition of aldolase (PMID: 36217034), a key glycolytic enzyme, increases the activation of AMPK α and the presence of G3BP1⁺ condensates (new Fig. 3A-E). The new data further show that the phenotype observed in MIN6-K8 cells exposed to resting glucose concentrations + pyruvate (a glycolytic metabolite downstream of aldolase) resembles the one observed in glucose stimulated MIN6-K8 cells with elevation of ATP levels and the resolution of G3BP1/2⁺ condensates, but not when cells were also co-treated with alogliptin. Taken together, these results imply that the rate of aldolase activity, conceivably by affecting the levels of cytosolic NADH, regulates oxidative phosphorylation and ATP production, while concomitantly modulating the AMPK α /mTOR activity ratio, the occurrence of G3BP1/2⁺/*Ins mRNA*⁺ condensates, and thereby insulin mRNA translation (new Fig. 8). As illustrated in the revised discussion, these findings are especially intriguing in view of the upregulation of aldolase B in islets of living donors with type 2 diabetes and in animal models of the disease.

3. Figure 3F: it seems that the figure contains the merge of two separate blots. If this is the

case it should be indicated. Comparison can be done only if some lanes on the same blot were spliced out.

The crops belonged to the same blot. However, now it is in Figure 4I and it has been changed to another blot from another replicate that is not cropped and it's from a single blot.

Reviewer #3

1) Based on Wigger et al. (2021) study, G3BP1 was significantly downregulated in the islets of living donors with T2D compared to normoglycemic individuals, while author found that the mRNA and protein almost have no change in the MIN6-K8 cells induced by glucose, why? Palmitate or other factors would affect G3BP1 expression? The expression level of G3BP1 in the islets of T2DM model mice need investigate?

We appreciate the interest of the reviewer for our previous findings in islets of living donors in relation to the findings presented in this manuscript. However, the fact that G3BP1 is downregulated in subjects with type 2 diabetes (T2D) doesn't necessarily imply that it should be acutely regulated by glucose, nor by palmitate, which also rapidly induced the resolution of G3BP1⁺ condensates. It is important to consider that patients with T2D are exposed to hyperglycemia, and most often hyperlipidemia, for months and years. Thus, it should not be expected that the outcome of the in vitro analyses presented in this manuscript phenocopy or fully explain the alterations identified in human islets of subjects affected by T2D. We emphasize that the scope of our paper was not to elucidate the reasons for downregulated expression of G3BP1 in T2D, but to gain insight first about its role in beta cells, and on such basis, elaborate hypotheses for how its downregulation in T2D could affect beta cell physiology. Testing such hypotheses, however, will be the aim of future studies.

2) Why only *ins1/2* is detectable in G3BP1+condensates of resting MIN6-K8 cells, while *Ptprn/ICA512*, *Pcsk1* and *Pcsk2* were not detected. The precise mechanism of G3BP1+ / and G3BP2+ /*Ins1/2* mRNA+ condensates formation needs to further research.

Apparently mRNAs encoding for secretory granule cargoes share several properties in common, including the binding to a common set of mRNA binding proteins and coordinated upregulation of translation in response to stimulation by glucose, but possibly other insulin secretagogues (Knoch et al 2004, PMID: 15039777, Knoch et al 2006 PMID: 16459313, Vasiljevic et al, <https://doi.org/10.1101/2021.05.07.443159>). However, it is not to be expected that their regulation is identical in every aspect. For instance, due to their lower levels compared to *insulin1/2* mRNA, *Ptprn/ICA512*, *Pcsk1* and *Pcsk2* mRNAs may not require sequestration in condensates for their translation to be suppressed in resting cells. Lack of detection of *Ptprn/ICA512*, *Pcsk1* and *Pcsk2* mRNAs in G3BP1/2⁺ condensates of resting MIN6-K8 cells may also be due to technical reasons, such as their probes for smRNA FISH not been equally effective or their levels being so much lower compared to *insulin1/2* mRNA. These negative results, on the other hand, do not negate the key novel outcome of our studies, that is the sequestration of *insulin1/2* mRNA in aldolase-regulated G3BP1/2⁺ condensates in resting beta cells.

3) More deep analysis of G3BP1+ / and G3BP2+ /Ins1/2 mRNA+ condensates in the islets of T2DM model mice (such as HFD, db/db and ob/ob) should be include.

As stated in our reply to point 1, this was not the aim of the present studies.

4) since different glucose level effect G3BP1+ / and G3BP2+ /Ins1/2 mRNA+ condensates, what is trend of G3BP1+ / and G3BP2+ /Ins1/2 mRNA+ condensates in the development of T2DM, please investigate it in the islets of mice.

As stated in the reply to point 1, this was not the aim of the present studies.

5) G3BP1 and G3BP2 are critical constituents of stress granules, and the mechanism is various (doi: 10.1016/j.cell.2020.03.046. doi: 10.1126/science.abf6548.). why author think G3BP1 and G3BP2 condensates regulated via AMPK α pathway in the islets, it seems too casual. Overall, the mechanism of ins1/2/G3BP1/G3BP2 cytosolic condensates formation is still confused. I would like to see a mRNA or protein analyses to show which genes are likely to be directly regulated by G3BP1 and G3BP2 in the MIN6-K8 cells.

We have now investigated more thoroughly the relationship between AMPK α , aldolase activity and G3BP1/2 condensates. We kindly refer to our reply to point in the minor comments of Reviewer #2

(6) Figure 2D showed that AMPK α was decreased, The relationship between G3BP1, and AMPK α is not clear and must be clarified.

See above

(7) Resting condition in Figure2-4 is 2.8 mM glucose. Resting condition in Figure 5 choose 2.8 mM and 3.3 mM glucose, please clarify the reason.

The reason for testing two different glucose concentrations in the case of mouse primary beta cells is that their threshold for glucose responsiveness is typically higher compared to insulinoma cell lines. The concentration of 3.3 mM glucose is the one routinely applied to study mouse primary beta cells in resting conditions.

(8) It is better to investigate this phenomenon that ins1/2/G3BP1/G3BP2 cytosolic condensates formation in the human β cell line (such as Endoc- β H1)

We provide evidence for the occurrence of G3BP1⁺ condensates in primary beta cells in situ of metabolically profiled normoglycemic living donors. Mechanistic investigations about such condensates in Endoc- β H1-5 could be the focus of future studies.

Minor comments

9) Please add the animal ethics committee in the methods of mice.

Thanks for the comments. This omission has been corrected.

10) As sample size for each experiment should be included. Also, details related to any replicates/repetitions can be included.

11) Quantifying the western blot images in Figure 3 and 2 would enhance the readability. The quantification of western blots in Fig. 2E is shown in Fig. 2F. As for the western blots for proinsulin and insulin in Fig. 3F (now Fig. 4I), we think that the quantifications of proinsulin and insulin by both ELISA and HTRF (as shown in Fig. S2B-C) are more reliable.

Dear Dr Solimena,

Thank you for submitting a revised version of your manuscript. Your study has now been seen by all original referees, who find that their previous concerns have been addressed and now recommend publication of the manuscript. There remain only a few minor edits requested by the referees and editorial points that have to be addressed before I can extend formal acceptance of the manuscript:

- Please reduce the number of keywords on the abstract page to five (ideally choosing broad general terms).
- Please rename the Conflict of Interest section into "Disclosure and Competing Interests Statement", in accordance with our updated Guide to Authors (<https://www.embopress.org/competing-interests>)
- As we are switching from a free-text author contribution statement towards a more formal statement based on Contributor Role Taxonomy (CRediT) terms, please remove the present Author Contribution section and instead specify each author's contribution(s) directly in the Author Information page of our submission system during upload of the final manuscript. See <https://casrai.org/credit/> for more information.
- Please adjust the in-text callouts for individual figures and figure panels:
there are callouts for Table 1-3 and also Table EV1-EV3, but there are only Tables 1-3 uplidd (although they should be renamed to Table EV1-EV3); there is a callout for Appendix Fig. S1, but no Appendix file uplidd; Fig. 7 has only one panel, so it shouldn't be labeled - label A should be removed from the figure and callouts.
- Please rename the the supplementary figures and tables, their legends as well as the callouts in the manuscript text (supplementary figures should be Figure EV1-EV6 and tables should be Tabela 1-3 should be Table EV1-EV3)
Legends should be removed from ms file and uploaded in the corresponding Excel file
- Please provide suggestions for a short 'blurb' text prefacing and summing up the conceptual aspect of the study in two sentences (max. 250 characters), followed by 3-5 one-sentence 'bullet points' with brief factual statements of key results of the paper; they will form the basis of an editor-written 'Synopsis' accompanying the online version of the article. Please also provide an altered synopsis image, making sure that the aspect ratio conforms to our website's format - it should be exactly 550 pixels wide and between 300-600 pixels high.
- Figure Legends (main + EV): Please note that the exact p values are not provided in the legends of figures 2B, C, F; 3A, C, D; 4A, E, F, G, H, M; 5B, C, D; supplementary figures 2A, B, D, E, G, L, 4C, D; 6B, C.
- Please adjust the order of the manuscript sections: Title page with complete author information, Abstract, Keywords, Introduction, Results, Discussion, Methods, Data Availability Section, Acknowledgements, Disclosure and Competing Interests Statement, References, Main figure legends, Tables, Expanded Figure Legends.

With best regards,

Cornelius Schneider

Cornelius Schneider, PhD
Editor
The EMBO Journal
c.schneider@embojournal.org

We realize that it is difficult to revise to a specific deadline. In the interest of protecting the conceptual advance provided by the work, we recommend a revision within 3 months (13th May 2025). Please discuss the revision progress ahead of this time with the editor if you require more time to complete the revisions. Use the link below to submit your revision:

Referee #1:

In this revised manuscript, the authors have adequately addressed our earlier comments. They have showed that insulin mRNA is translationally repressed in resting beta cells and stored in G3BP granules. They also demonstrate that G3BP1 and 2 play different roles in the lifecycle of those mRNAs by a not yet determined mechanism. I think these findings are of interest to the field. I am supportive of publication with some final, minor modifications:

- 1) The actinomycin D data in figures 4F and 4G are more appropriately presented as a decay curve instead of raw Ct values. This will be confusing to the reader.
- 2) Concerning figure 3B, I found it notable that the mRNA signal in the last two panels looked more diffuse despite the presence of G3BP granules. It would be interesting to quantify that signal as well as the G3BP signal.
- 3) The subsection title on page 7 " Glucose stimulated G3BP1+ condensate dissolution its aldolase and ATP dependent" contains a typo since "its" should be "is".

Referee #2:

In this revised version, the authors have addressed most of previously raised critical points and also added data that help elucidate the signalling cascade implicated.

Minor comment:

The supplementary figures are mentioned in the text without a logical order. In addition, I could not find any reference in the text about the experiments illustrated in Figure S2I-M. It would be helpful to carefully check what is shown in the supplementary Figures, clearly cite them in the text and label them in the order in which they appear in the text.

Point-by-point response

Referee #1:

In this revised manuscript, the authors have adequately addressed our earlier comments. They have showed that insulin mRNA is translationally repressed in resting beta cells and stored in G3BP granules. They also demonstrate that G3BP1 and 2 play different roles in the lifecycle of those mRNAs by a not yet determined mechanism. I think these findings are of interest to the field. I am supportive of publication with some final, minor modifications:

1) The actinomycin D data in figures 4F and 4G are more appropriately presented as a decay curve instead of raw Ct values. This will be confusing to the reader.

Figure 4F and 4G have been replaced with a decay curve as requested.

2) Concerning figure 3B, I found it notable that the mRNA signal in the last two panels looked more diffuse despite the presence of G3BP granules. It would be interesting to quantify that signal as well as the G3BP signal.

We are thankful for the observation. We have added a new Figure 3F showing the fluorescence intensity of *Ins1/2* mRNA located to G3BP1⁺ condensates normalized to the cytosolic *Ins1/2* mRNA signal. These data do not indicate differences between the tested conditions.

3) The subsection title on page 7" Glucose stimulated G3BP1+ condensate dissolution its aldolase and ATP dependent" contains a typo since "its" should be "is".

Corrected

Referee #2:

In this revised version, the authors have addressed most of previously raised critical points and also added data that help elucidate the signalling cascade implicated.

Minor comment:

The supplementary figures are mentioned in the text without a logical order. In addition, I could not find any reference in the text about the experiments illustrated in Figure S2I-M. It would be helpful to carefully check what is shown in the supplementary Figures, clearly cite them in the text and label them in the order in which they appear in the text.

We thank the reviewer for this careful remark, which is correct. As we inadvertently omitted to refer to those supplementary data in the manuscript. This mistake has now been corrected.

Dear Dr. Solimena,

I am pleased to inform you that your manuscript has been accepted for publication in the EMBO Journal.

Yours sincerely,

Cornelius Schneider, PhD
Editor
The EMBO Journal
c.schneider@embojournal.org
